# Transcriptomic analyses reveal rhythmic and CLOCK-driven pathways in human skeletal muscle

Laurent Perrin[1,2,3,4], Ursula Loizides-Mangold[1,2,3,4†], Stéphanie Chanon[5†], Cédric Gobet[6,7†], Nicolas Hulo[4,8], Laura Isenegger[8], Benjamin D Weger[6], Eugenia Migliavacca[6], Aline Charpagne[6], James A Betts[9], Jean-Philippe Walhin[9], Iain Templeman[9], Keith Stokes[9], Dylan Thompson[9], Kostas Tsintzas[10], Maud Robert[11], Cedric Howald[4,12], Howard Riezman[13], Jerome N Feige[6,7], Leonidas G Karagounis[14,15], Jonathan D Johnston[16], Emmanouil T Dermitzakis[4,12], Frédéric Gachon[6,7‡], Etienne Lefai[5‡], Charna Dibner[1,2,3,4*]

[1]Division of Endocrinology, Diabetes, Hypertension and Nutrition, Department of Internal Medicine Specialties, University Hospital of Geneva, Geneva, Switzerland; [2]Department of Cell Physiology and Metabolism, Faculty of Medicine, University of Geneva, Geneva, Switzerland; [3]Diabetes Center, Faculty of Medicine, University of Geneva, Geneva, Switzerland; [4]Institute of Genetics and Genomics of Geneva, Geneva, Switzerland; [5]CarMeN Laboratory, INSERM U1060, Oullins, France; [6]Nestlé Institute of Health Sciences, Lausanne, Switzerland; [7]School of Life Sciences, Ecole Polytechnique Fédérale de Lausanne, Lausanne, Switzerland; [8]Service for Biomathematical and Biostatistical Analyses, Institute of Genetics and Genomics in Geneva, University of Geneva, Geneva, Switzerland; [9]Department for Health, University of Bath, Bath, United Kingdom; [10]MRC/ARUK Centre for Musculoskeletal Ageing, School of Life Sciences, University of Nottingham, Nottingham, United Kingdom; [11]Department of Digestive and Bariatric Surgery, Edouard Herriot University Hospital, Lyon, France; [12]Department of Genetic Medicine and Development, Faculty of Medicine, University of Geneva, Geneva, Switzerland; [13]Department of Biochemistry, NCCR Chemical Biology, University of Geneva, Geneva, Switzerland; [14]Experimental Myology and Integrative Biology Research Cluster, Faculty of Sport and Health Sciences, University of St Mark and St John, Plymouth, United Kingdom; [15]Institute of Nutritional Science, Nestlé Research Centre, Lausanne, Switzerland; [16]Faculty of Health and Medical Sciences, University of Surrey, Guildford, United Kingdom

*For correspondence:
Charna.dibner@hcuge.ch

†These authors contributed equally to this work
‡These authors also contributed equally to this work

**Abstract** Circadian regulation of transcriptional processes has a broad impact on cell metabolism. Here, we compared the diurnal transcriptome of human skeletal muscle conducted on serial muscle biopsies in vivo with profiles of human skeletal myotubes synchronized in vitro. More extensive rhythmic transcription was observed in human skeletal muscle compared to in vitro cell culture as a large part of the in vivo mRNA rhythmicity was lost in vitro. siRNA-mediated clock disruption in primary myotubes significantly affected the expression of ~8% of all genes, with impact on glucose homeostasis and lipid metabolism. Genes involved in GLUT4 expression, translocation and recycling were negatively affected, whereas lipid metabolic genes were altered to promote activation of lipid utilization. Moreover, basal and insulin-stimulated glucose uptake were significantly reduced upon *CLOCK* depletion. Our findings suggest an essential role for the circadian coordination of skeletal muscle glucose homeostasis and lipid metabolism in humans.

DOI: https://doi.org/10.7554/eLife.34114.001

## Introduction

Circadian rhythms are daily cycles of most bodily processes driven by a system of intrinsic biological clocks organized in a complex hierarchical manner. This mechanism ensures a temporal coordination of physiology and behavior with a near 24 hr period of rest-activity and feeding-fasting cycles, thus providing the organism with an evolutionary conserved advantage (*Albrecht, 2012*; *Spoelstra et al., 2016*). In mammals, the circadian system is driven by a central pacemaker, situated in the paired suprachiasmatic nuclei (SCN) of the hypothalamus, and by secondary oscillators located in peripheral organs. The SCN clock is readjusted on a daily basis, mainly by light inputs coming from the retina. In turn, the central pacemaker orchestrates peripheral clocks through a combination of neuronal, endocrine, and metabolic signaling pathways (*Saini et al., 2015*). As a result, metabolic processes in the liver, skeletal muscle, and other organs are subject to daily oscillations (*Asher and Sassone-Corsi, 2015*) with the SCN keeping these rhythms in appropriate synchrony with each other.

Skeletal muscle is responsible for ~70% of glucose uptake resulting from ingested carbohydrates (*DeFronzo et al., 1981*; *Gachon et al., 2017*), and perturbations in glucose sensing and metabolism in this organ are strongly associated with insulin resistance in type 2 diabetes (T2D) (*Muoio and Newgard, 2008*). In rodents, it has been established that the skeletal muscle clock plays an essential role in maintaining proper metabolic homeostasis, with skeletal muscle pathologies stemming from clock disruption via deletion of the core clock component *Bmal1* (*Andrews et al., 2010*; *Harfmann et al., 2015*). Disruption of muscle insulin sensitivity by modulating glucose uptake, with a reduction in *Glut4* mRNA and protein levels, has been reported in two muscle-specific *Bmal1* knock-out (KO) models (*Dyar et al., 2014*; *Harfmann et al., 2016*). In humans, diurnal variations in glucose tolerance have been described (*Kalsbeek et al., 2014*), although the molecular mechanism responsible for such variations remains largely unexplored. Feeding-fasting cycles accompanying rest-activity rhythms represent major timing cues in the synchronization of peripheral clocks, including skeletal muscle oscillators (*Dibner and Schibler, 2015*; *Wehrens et al., 2017*). Although several studies have reported that in murine models 3.4% to 16% of the skeletal muscle transcriptome is expressed in a circadian manner (*McCarthy et al., 2007*; *Miller et al., 2007*; *Dyar et al., 2014*; *Zhang et al., 2014*; *Hodge et al., 2015*), it is unclear to what extent the muscle circadian transcriptome is regulated by feeding-fasting rhythms and additional central synchronizers, and how local muscle clocks contribute in generating these transcript oscillations. Cell-autonomous circadian clocks operating in human primary skeletal myotubes (hSKM) established from muscle biopsies have been recently characterized (*Perrin et al., 2015*; *Hansen et al., 2016*; *Loizides-Mangold et al., 2016*). Importantly, proper function of these cellular oscillators is indispensable for the normal secretion of interleukin 6 (IL-6), interleukin-8 (IL-8), the monocyte chemotactic protein 1 (MCP-1) and additional myokines, regulating skeletal muscle insulin sensitivity and inflammation (*Perrin et al., 2015*).

In order to dissect the impact of the endogenous circadian clock on skeletal muscle gene transcription from external factors and their reciprocal influence, we performed a genome-wide transcriptome analysis by high-throughput RNA sequencing (RNA-seq) in skeletal muscle biopsies collected from human subjects placed under a controlled laboratory routine, as well as in cultured hSKM synchronized in vitro in the presence of a functional or compromised clock. An important overlap between genes exhibiting rhythmic patterns in tissue biopsies and in synchronized hSKM was observed. Expression patterns of genes related to insulin response, myokine secretion, and lipid metabolism were strongly altered in the absence of a fully functional clock in vitro. These transcriptional changes had important functional outputs, with basal as well as insulin-stimulated glucose uptake and lipid metabolism being altered by perturbation of the oscillators operative in hSKM. Altogether, these results strongly suggest that cell-autonomous skeletal muscle clocks drive rhythmic gene expression and are indispensable for proper insulin response, lipid homeostasis, and myokine secretion by the skeletal muscle.

## Results

### Diurnal rhythms of gene expression in human skeletal muscle under controlled laboratory routine

To assess rhythms of gene expression in human skeletal muscle, RNA samples derived from *vastus lateralis* biopsies taken every 4 hr across 24 hr from 10 healthy volunteers were analyzed by total RNA-seq (see *Supplementary file 1*-table S1 for in vivo donor characteristics, n = 10). Sample collection was performed under controlled laboratory routine by implementing a protocol designed to minimize the effect of confounding factors (see Materials and methods and [*Loizides-Mangold et al., 2017*]). In total 13,377 genes were quantified at the exonic level (*Figure 1—source data 1*), of which 9211 genes were quantified at the intronic level as well. To identify genes with coordinated rhythmic expression, we used a mixed linear model with harmonic terms across the 10 individuals at the pre-mRNA (intronic signal) and mRNA (exonic signal) levels. This method allowed for the identification of 5748 rhythmic genes that were rhythmic at the pre-mRNA or/and mRNA level with a False Discovery Rate (FDR) of 5% (*Figure 1A*). When rhythmicity levels were further analyzed, it became apparent that 4792 genes showed rhythmic transcription at the intronic pre-mRNA level (*Figure 1A*, upper and middle left panels). However, from these rhythmic pre-mRNA transcripts, only 57% were also rhythmic at the mRNA level (R-I.R-E, upper panels *Figure 1A*), likely because of the longer half-life of their mRNA. Indeed, approximation of mRNA half-life by the exon/intron ratio showed that among these rhythmic pre-mRNA transcripts, those that are only rhythmic at the pre-mRNAs level (R-I) have a longer half-life compared to those that are rhythmic at the mRNA level (R-E and R-I.R-E, *Figure 1B*). The R-I.R-E group of genes was enriched in circadian clock genes and genes encoding RNA-binding proteins, whereas the R-I group was enriched for genes encoding proteins involved in mRNA translation, mitochondrial activity, TCA cycle, and lipid metabolism (*Figure 1—source data 2*). In parallel, around 10% of the quantified transcriptome (956 genes) were only rhythmic at the mRNA level (R-E, *Figure 1A*, lower panels), likely through post-transcriptional regulation and in particular mRNA degradation (*LuckLück et al., 2014*; *Wang et al., 2018*). This group was enriched in genes encoding proteins involved in ribosome biogenesis and protein transport (*Figure 1—source data 2*). Amplitude distribution suggested that among the genes that were rhythmic at the pre-mRNA level, those with higher amplitude of transcription had a greater probability to be rhythmic at the mRNA accumulation level (R-I.R-E, *Figure 1C*). Taken together, our results highlight the high rhythmicity of gene expression in human skeletal muscle, even under controlled laboratory routine. However, the number of genes being qualified as significantly rhythmic at the pre-mRNA and/or mRNA level was strongly dependent on the threshold level that was applied (*Figure 1D*).

As previously reported in mice, rhythmic gene transcription was distributed into two phases of transcript accumulation (04:00 and 16:00, *Figure 1E*). The afternoon peak (16:00) was enriched in genes related to muscle contraction and mitochondrial activity (*Figure 1—source data 2*), whereas homologous genes in rodents were shared between the active but also the resting phase (*McCarthy et al., 2007*; *Miller et al., 2007*; *Hodge et al., 2015*). In contrast, among the genes highly activated in the middle of the night (04:00), many were associated with inflammation and immune response (*Figure 1F*).

Among the rhythmic genes, we observed high amplitude oscillations for the core clock genes *ARNTL* (*BMAL1*), *NPAS2*, *CLOCK*, *PER2*, *PER3*, *CRY2*, *NR1D1* (*REV-ERBα*) and *RORA*, which were well synchronized among the 10 individuals (*Figure 1G*). In addition, several transcription factors associated with muscle metabolism and physiology like *TFEB*, a key regulator of lysosomal biogenesis and autophagy that also regulates glucose homeostasis and oxidative phosphorylation (*Mansueto et al., 2017*), *KLF13*, which plays a role in cardiac muscle cell development (*LavalleeLavallée et al., 2006*), and *KLF15*, an important transcriptional regulator of muscle lipid metabolism (*Haldar et al., 2012*), showed an oscillatory profile. Moreover, also *PPARD*, the most abundant PPAR isoform in skeletal muscle and master regulator of muscle mitochondrial function (*Jordan et al., 2017*), and *MYOD1*, the key regulator of myogenesis and direct target of BMAL/CLOCK (*Andrews et al., 2010*) were rhythmically expressed in human skeletal muscle, likely orchestrating the temporal muscle transcriptome (*Figure 1H*). In addition to these transcription factors, genes involved in the secretion of myokines, glucose homeostasis and lipid metabolism displayed rhythmic transcription (*Figure 1—figure supplement 1A,B and C*, respectively).

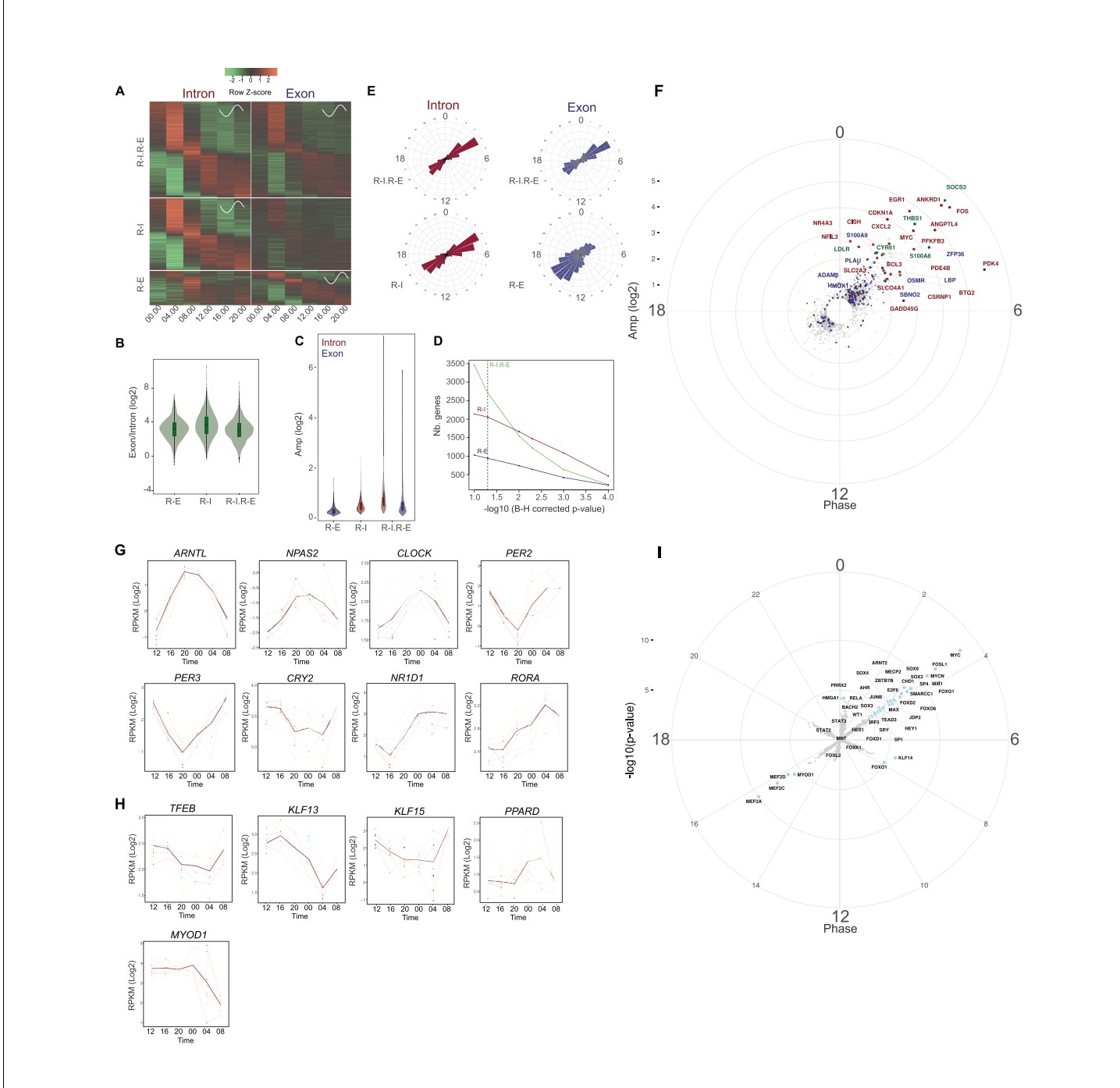

**Figure 1.** Rhythmic gene expression in human skeletal muscle. (**A**) Heat map showing genes rhythmic at the pre-mRNA and mRNA level (R-I.R-E: upper panel), at the pre-mRNA level only (R-I: middle panel), and at the mRNA level only (R-E: lower panel). Standardized relative gene expression is indicated in green (low) and red (high) and ordered by their respective phase. (**B**) mRNA half-life proxy by exon/intron ratio showing lower stability for genes with rhythmic mRNA (**R–E**) profiles. (**C**) Amplitude distribution of genes that are rhythmic only at the mRNA level (R-E, blue), the pre-mRNA level (R-I, red), or rhythmic for both (R-I.R-E). Genes with higher amplitude of transcription at the pre-mRNA level have a higher probability to be rhythmic at the mRNA level (R-I.R-E). (**D**) Number of genes in each group in relation to the -$\log_{10}$ BH corrected p-value; dashed line indicates threshold of 0.05. (**E**) Phase distribution at the pre-mRNA and mRNA level for the three groups described in (**A**). (**F**) Phase distribution for genes activated by acute muscle exercise (red), inflammation (blue), or both (green). (**G**) Temporal expression of core clock components, and (**H**) key muscle transcription factors. N = 10 human muscle biopsy donors. (**I**) Phase distribution of predicted rhythmic DNA motif activity.

DOI: https://doi.org/10.7554/eLife.34114.002

*Figure 1 continued on next page*

*Figure 1 continued*

The following source data and figure supplement are available for figure 1:

**Source data 1.** List of 9211 genes identified by RNA-seq analysis in human skeletal muscle.
DOI: https://doi.org/10.7554/eLife.34114.004
**Source data 2.** GO term enrichment analysis for transcripts identified as rhythmic in human skeletal muscle.
DOI: https://doi.org/10.7554/eLife.34114.005
**Figure supplement 1.** Temporal gene expression in human skeletal muscle.
DOI: https://doi.org/10.7554/eLife.34114.003

To gain more insight into the rhythmic transcriptional regulation observed in this dataset, we performed a DNA-binding motif enrichment analysis to identify those with rhythmic activity. As shown in *Figure 1I*, the 16:00 peak is enriched in MEF2 and MYOD1 motifs, in phase with *MYOD1* expression, and both proteins synergize to activate gene expression (*Taylor and Hughes, 2017*). In parallel, the 4:00 peak is enriched in MYC and AP1 families of transcription factors, both downstream of the MAP kinase pathway activated by exercise (*Aronson et al., 1997*) or wound-induced inflammation (*Aronson et al., 1998*).

## Cell-autonomous circadian clocks regulate functional gene expression in hSKM

To assess the impact of cell-autonomous circadian clocks operative in hSKM on skeletal muscle gene transcription and function, we used our previously developed model of efficient siCLOCK-mediated clock disruption (*Perrin et al., 2015*). RNA-seq was conducted on siCLOCK-transfected hSKM obtained from two male donors matched for age and BMI (*Supplementary file 1*-table S1, donors 1 and 2, in vitro part). Human primary myoblasts were cultured and differentiated in vitro into myotubes, transfected with siRNA, synchronized in vitro with a forskolin pulse, with subsequent sample collection every 2 hr during 48 hr for RNA-seq analysis (*Figure 2—figure supplement 1* and Materials and methods). *CLOCK* expression was reduced by at least 80% upon siCLOCK depletion, as assessed by RNA-seq and quantitative real-time PCR (*Figure 2A* and *Figure 2—figure supplement 2A*). In parallel, bioluminescence profiles derived from hSKM transduced with a lentiviral *Bmal1*-luciferase vector were monitored as described previously (*Perrin et al., 2015*). As expected, the circadian amplitude of *Bmal1*-luciferase bioluminescence reporter profile became dampened upon siCLOCK compared to siControl and non-transfected counterparts (*Figure 2—figure supplement 2B*).

We first performed a differential analysis of the global gene expression profile across all 25 time points, starting from 0 hr and until 48 hr following synchronization. Out of the 16,776 mapped genes, the median values for all the time points, reflecting the overall expression levels of 1330 genes (8%), were significantly altered in siCLOCK-transfected hSKM compared to their control counterparts, with 742 being downregulated and 588 being upregulated (*Figure 2B*; *Figure 2—source data 1*). As expected, core clock gene expression was affected, with *NR1D1* (*REVERBα*), *NR1D2* (*REVERBβ*), *PER3*, *TEF*, *BHLHE41* (*DEC2*), and *DBP* being significantly downregulated, and *CRY1* being upregulated upon *CLOCK* depletion (*Figure 2A*).

## Functional clocks operative in hSKM are required for proper lipid metabolism and response to insulin

Remarkably, genes encoding for proteins essential for vesicle formation including SNAREs, solute transporters, and Rab GTPases exhibited significantly altered expression levels upon *CLOCK* depletion (*Supplementary file 1*-table S2). Additional genes involved in secretion pathways and exhibiting altered mRNA expression levels upon *CLOCK* depletion are listed in *Supplementary file 1*-table S3. Using the Panther classification system (*Mi et al., 2017*) for gene ontology (GO) term analysis, overrepresentation of genes associated with the regulation of nucleotide metabolism, transcription and RNA processing, as well as muscle contraction were identified within the significantly down- and/or upregulated genes (*Supplementary file 1*-table S4, and *Figure 2—source data 2*). Furthermore, enrichment analysis using the Reactome pathway database was performed on the down- and/or upregulated genes. Of note, overrepresentation of genes related to muscle contraction, regulation

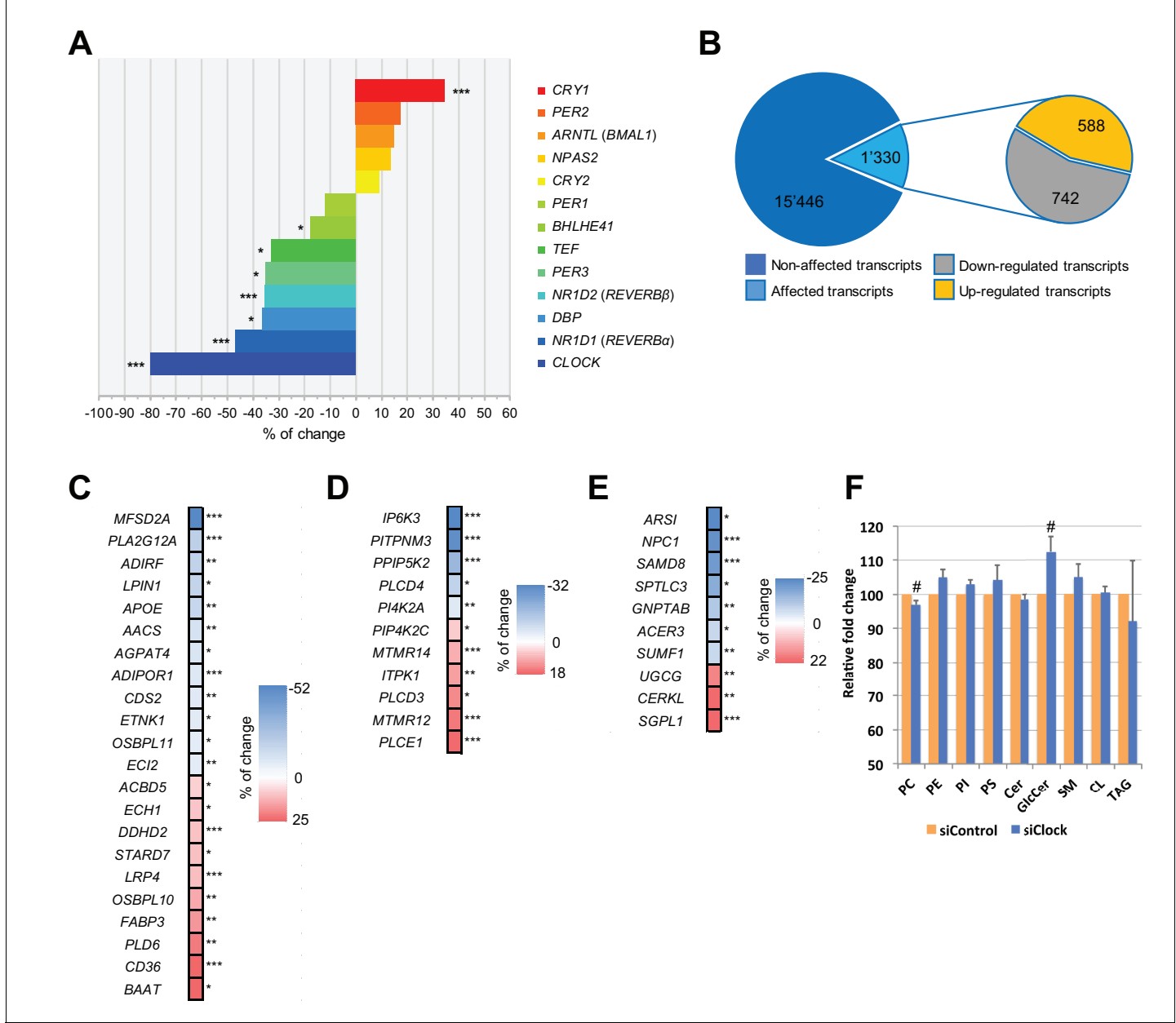

**Figure 2.** Disruption of the circadian oscillator impacts on functional gene expression hSKM. Differential gene expression analysis between hSKM bearing a disrupted (siCLOCK) or intact (siControl) circadian clock. Comparison of the median gene expression at all analyzed circadian time points between the two groups. A total of 16,776 genes were detected by RNA-seq as expressed above the threshold of counts per million (CPM) >3. (A) Core clock genes; (B) 15,446 genes remained unchanged (dark blue), and 1330 genes exhibited a significantly different expression level upon siCLOCK-mediated clock disruption (light blue), with 588 being up-regulated (orange) and 742 down-regulated (grey) (FDR <0.05 and p-value <0.05). Altered genes comprised those related to glycerophospholipid and triglyceride metabolism, storage and transport (C) inositol phosphate metabolism (D) and sphingolipid metabolism and storage (E). (F) Relative levels of PC, PE, PI, PS, Cer GlcCer, SM, CL and TAG, analyzed by mass spectrometry based lipidomics in hSKM cells transfected with either siControl (orange bar) or siCLOCK (blue bar). Total phosphatidylcholine (PC) and glycosylceramide (GlcCer) levels are significantly decreased or increased, respectively, upon siCLOCK transfection. Data are mean ± SEM, N = 4 (# p-value <0.05). (*) for FDR <0.05, (**) for FDR <0.01, (***) for FDR <0.001.

DOI: https://doi.org/10.7554/eLife.34114.006

The following source data and figure supplements are available for figure 2:

**Source data 1.** List of 16,776 genes identified in hSKM by RNA-seq and used for the differential analysis.

DOI: https://doi.org/10.7554/eLife.34114.009

*Figure 2 continued on next page*

Figure 2 continued

**Source data 2.** GO term enrichment analysis, using the Panther classification system, for transcripts that were down- and/or upregulated upon clock disruption.
DOI: https://doi.org/10.7554/eLife.34114.010

**Source data 3.** Reactome pathway analysis, using the Panther classification system for transcripts that were down- and/or upregulated upon clock disruption.
DOI: https://doi.org/10.7554/eLife.34114.011

**Figure supplement 1.** Study design.
DOI: https://doi.org/10.7554/eLife.34114.007

**Figure supplement 2.** siRNA-mediated *CLOCK* knockdown leads to a flattening of the *Bmal1-luc* circadian oscillation amplitude in hSKM.
DOI: https://doi.org/10.7554/eLife.34114.008

of gene transcription, and cellular responses to stress and membrane trafficking were also identified (*Supplementary file 1*-table S5, *Figure 2—source data 3*).

In addition, 42 transcripts involved in lipid metabolism were affected by *CLOCK* disruption. These comprised genes related to glycerophospholipid and triglyceride metabolism as well as lipid storage and transport (*Figure 2C*), in addition to those regulating inositol phosphate (*Figure 2D*) and sphingolipid metabolic pathways (*Figure 2E*). Importantly, the observed modifications in gene expression level were in an accord with significant alterations in absolute lipid metabolite levels, resulting in total phosphatidylcholine levels being downregulated, and glycosylceramide levels being upregulated in the absence of a functional myotube clock (*Figure 2F*, *Supplementary file 1*-table S1, donors 7–10 for in vitro part). The first matching a reduction in lysophosphatidylcholine symporter 1 (*MFSD2A*) and phosphatase *LPIN1* levels (*Figure 2C and F*), and the latter matching the transcriptome outcome for UDP-glucose ceramide glucosyltransferase (*UGCG*) the key enzyme of de novo glucosylceramide biosynthesis (*Figure 2E and F*).

Our differential analysis in human muscle cells demonstrates that genes involved in insulin-stimulated and contraction-induced glucose uptake, comprising *TBC1D13*, T*BC1D4* (*AS160*), *14-3-3θ* (*YWHAQ*), *RAB35*, *STX6,* and *PDPK1* (*PDK1*), were significantly downregulated upon siCLOCK (*Supplementary file 1*-table S2), highlighting the pleiotropic effect of the skeletal muscle CLOCK gene/protein in regulating glucose uptake in response to insulin and/or to contraction.

To determine the protein levels of candidate genes identified by RNA-seq, hSKM cells established from five additional donor biopsies (for donor characteristics see *Supplementary file 1*-table S1) were transfected by siRNA targeting *CLOCK*. Matching the changes observed by RNA-seq, *CLOCK* mRNA was reduced by siCLOCK as determined by RT-qPCR (*Figure 3A*), leading to a reduction in CLOCK protein expression by 74% (*Figure 3B*). Moreover, the expression of the 14-3-3θ protein, a key regulator of GLUT4 translocation (*Sakamoto and Holman, 2008*; *Kleppe et al., 2011*), was decreased by about 28% under these conditions (*Figure 3C*), matching the decrease in its gene expression (*Supplementary File 1*-table S2). In contrast, AS160 protein levels did not change significantly (*Figure 3—figure supplement 1*) despite a reduction at the mRNA expression level (*Supplementary file 1*-table S2).

Finally, we analyzed the impact of clock disruption on the ability of hSKM to take up glucose in response to insulin. The assessment of glucose uptake, using a radioactive glucose analogue, demonstrated an increase in glucose uptake upon insulin stimulation in non-synchronized siControl and siCLOCK-transfected myotubes (siControl: basal *vs.* insulin p-value = 0.019; siCLOCK: basal *vs.* insulin p-value = 0.017). Importantly, we observed a significant decrease in both basal (30%), and insulin-stimulated (27%) glucose uptake in siCLOCK-transfected myotubes, as compared to their siControl counterparts (*Figure 3D* left and right panels, respectively).

## RNA-seq reveals rhythmically expressed genes in cultured hSKM synchronized in vitro

We next aimed at identifying genes that exhibited rhythmic profiles in hSKM synchronized in vitro. The existing algorithms JTK_CYCLE (*Hughes et al., 2010*) and CosinorJ (*Mannic et al., 2013*) do not allow for a satisfactory analysis of datasets containing large differences in amplitude, observed among the two cycles in our datasets. We therefore developed a novel algorithm, adapted for the analysis of our RNA-seq datasets, comprising high-resolution analysis of two full cycles (25 time

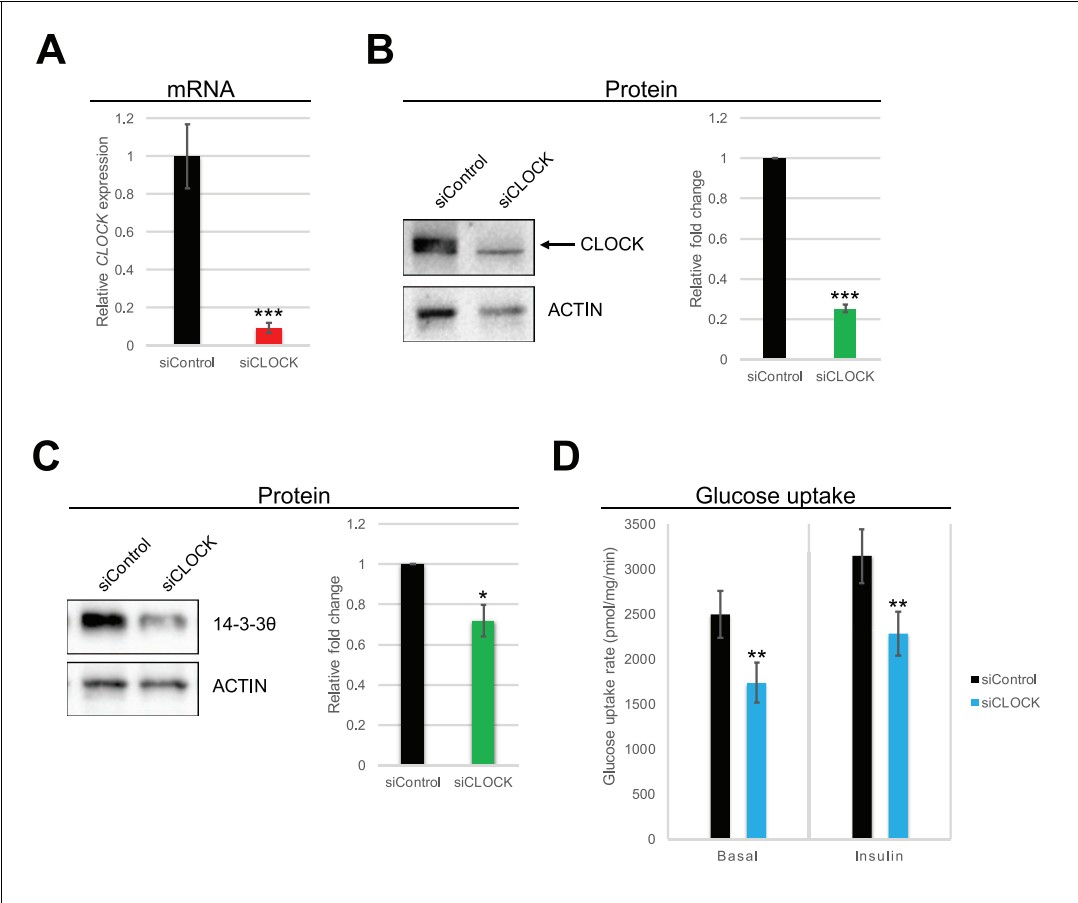

**Figure 3.** Basal and insulin-induced glucose uptake by hSKM is downregulated in the absence of a functional circadian clock. (**A**) *CLOCK* mRNA was measured in hSKM cells transfected with siControl or siCLOCK by RT-qPCR and normalized to the mean of *9S* and *HPRT*. *CLOCK* expression was reduced by 91 ± 2% (mean ± SEM, N = 4; (***) p-value <0.001) in siCLOCK-transfected cells. Protein levels of CLOCK (**B**) and 14-3-3θ (**C**) were assessed by western blot. Left panel: representative western blot; right panel protein quantification over all monoplicates (mean ± SEM, N = 5). CLOCK and 14-3-3θ protein levels were reduced by 75 ± 2%, and 28 ± 8%, respectively. (**D**) Glucose uptake rates (in pmol/mg.min) measured in the absence (basal) or presence (insulin) of insulin (1 hr, 100 nM) in siControl or siCLOCK-transfected cells. Note significant reduction of basal (31 ± 3%) and insulin-stimulated glucose uptake (28 ± 3%). Data are mean ± SEM of four independent experiments using myotubes from four different donors (same as for A-C). (*) p-value <0.05, (**) p-value <0.01, (***) p-value <0.001.

DOI: https://doi.org/10.7554/eLife.34114.012

The following figure supplement is available for figure 3:

**Figure supplement 1.** TBC1D4/AS160 protein levels are not affected by siCLOCK.

DOI: https://doi.org/10.7554/eLife.34114.013

points over 48 hr) following in vitro synchronization (see Materials and methods for details). Our in vitro algorithm was validated on the temporal expression profiles of key core clock genes from two published large-scale time series (*Atger et al., 2015*; *Petrenko et al., 2016*) and on the model data-set cycMouseLiverRNA from the MetaCycle R package. Briefly, after conversion of the raw data to $\log_2$ RPKM values and filtering for $\log_2$ RPKM >0, temporal patterns of the resulting 12,985 genes were grouped into 16 models (*Figure 4A*, *Figure 4—source datas 1* and *2*), with models 1 to 15 comprising 994 rhythmic genes (7.65%), and model 16 comprising 11,991 non-rhythmic genes (92.35%).

Because the number of rhythmic genes exhibited larger variations between the two analyzed cell cultures, established from two different human individuals than between siControl and siCLOCK (*Figure 4—figure supplement 1A*), we proceeded with a deeper analysis of models 1 to 4, which group together genes that are rhythmic in the siControl situation for both donor cell cultures. According to our analysis, model 1 comprises 73 genes, classified as rhythmic in both donors upon

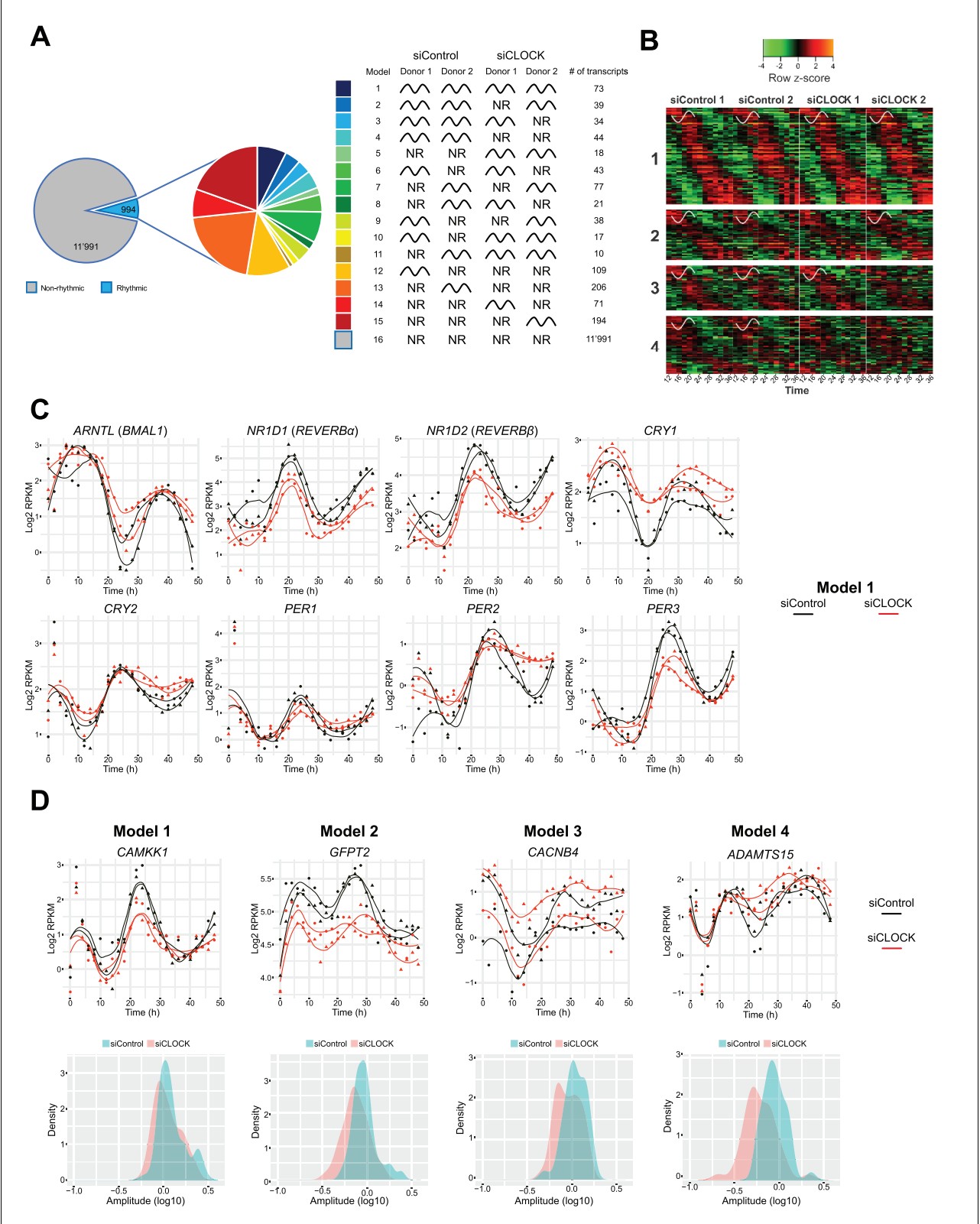

**Figure 4.** Temporal gene expression analysis in human skeletal myotubes bearing a disrupted or functional circadian clock. (**A**) Total of 12,985 genes were identified by RNA-seq as expressed above log$_2$ RPKM >0. Genes were subjected to the circadian analysis, adapted for high-resolution datasets over 48 hr. Genes were categorized as rhythmic or non-rhythmic (NR) (left diagram) and rhythmic genes (994) were grouped into 15 models (right panel). Genes that were non-rhythmic in either one of the 15 models (11,991 genes) are represented in model 16. (**B**) Heat maps for genes assigned to

*Figure 4 continued on next page*

eLIFE Research article                                                                                         Human Biology and Medicine

*Figure 4 continued*

models 1 to 4. Relative expression is indicated in green (low) and red (high). (**C**) Individual temporal expression profiles of core clock genes *ARNTL*, *NR1D1*, *NR1D2*, *CRY1*, *CRY2*, *PER1*, *PER2* and *PER3* in siControl or siCLOCK-transfected cells. (**D**) Upper panel: Representative examples for genes assigned to models 1–4. Lower panel: Circadian amplitude quantification of siControl and siCLOCK in models 1–4.

DOI: https://doi.org/10.7554/eLife.34114.014

The following source data and figure supplements are available for figure 4:

**Source data 1.** List of genes found in each of the 16 models identified by rhythmic analysis of the RNA-seq data.

DOI: https://doi.org/10.7554/eLife.34114.018

**Source data 2.** This dataset contains the $\log_2$ RPKM values for all 25 time points (0 to 48 hr) and the mean of all time points per donor and per condition (siControl/siCLOCK) as well as the model where each gene is grouped.

DOI: https://doi.org/10.7554/eLife.34114.019

**Source data 3.** GO term enrichment analysis, using the Panther classification system, for transcripts that were grouped into model 1.

DOI: https://doi.org/10.7554/eLife.34114.020

**Source data 4.** Reactome enrichment analysis, using the Panther classification system, for genes that were grouped into model 1.

DOI: https://doi.org/10.7554/eLife.34114.021

**Figure supplement 1.** Comparison of rhythmic transcript profiles between the two donors.

DOI: https://doi.org/10.7554/eLife.34114.015

**Figure supplement 2.** Temporal profiles of core clock transcript expression.

DOI: https://doi.org/10.7554/eLife.34114.016

**Figure supplement 3.** Genes involved in cell cycle regulation exhibit circadian expression profile in hSKM.

DOI: https://doi.org/10.7554/eLife.34114.017

siControl and siCLOCK, models 2 and 3 include 39 and 34 genes, respectively, that are rhythmic in both siControl donors and in one siCLOCK donor respectively, and model 4 comprises 44 genes that are rhythmic in siControl, but not in siCLOCK (*Figure 4B*). Circadian core clock genes clustered to model 1, as they exhibited a rhythmic mRNA profile in siControl and a flattened, but still rhythmic, profile upon siCLOCK (*Figure 4C* and *Figure 4—figure supplement 2*), indicating a presence of partially functional circadian oscillator. Of note, classification of a temporal gene profile as rhythmic by our algorithm did not take into consideration amplitude alterations, like those generated by siCLOCK-treatment, as long as the temporal profile was qualified as circadian. As amplitude values were indeed often blunted upon siCLOCK treatment, to quantify such amplitude changes a $\log_{10}$ transformation was applied, providing approximation to a normal distribution using a paired *t*-test. In agreement with our previous publication (*Perrin et al., 2015*), the amplitude of mRNA accumulation was significantly decreased in siCLOCK samples (*Supplementary file 1*-table S6, *Figure 4—figure supplement 1B*).

In summary, 190 genes were qualified as rhythmic in the two analyzed cell cultures, and were clustered into models 1–4 (*Figure 4—source datas 1* and *2*), as exemplified in *Figure 4D* (upper panels) and in *Figure 4—figure supplement 1C*. Importantly, similarly to core clock genes, also these functional genes exhibited a blunted circadian amplitude upon clock disruption (*Figure 4D*, lower panels, *Figure 4—figure supplement 1B*). For instance, *CAMKK1*, classified as rhythmic in both siControl and siCLOCK conditions (model 1), exhibited a significant circadian amplitude reduction upon siCLOCK (*Figure 4D*, *Supplementary file 1*-table S6). In addition, *SERPINE1*, a myokine whose secretion by myotube cells was reduced upon clock disruption (*Perrin et al., 2015*), presented lower amplitude in siCLOCK-transfected cells (*Supplementary File 1*-table S6). Panther database analysis for genes assigned to models 1–4 suggested enrichment for a number of GO term and Reactome pathways, comprising cell cycle and mitotic regulation (*Figure 4—figure supplement 3*, *Supplementary file 1*-tables S4-5, *Figure 4—source datas 3* and *4*).

## Comparative analysis of diurnal rhythms of gene expression in human skeletal muscle tissue and cultured hSKM

Consequently, we compared rhythmic gene expression between muscle biopsy and cultured hSKM cells (*Figure 5A*). Among the 190 transcripts that were identified as rhythmic in hSKM cells (*Figure 4A*, models 1–4, *Figure 5—source data 1*), 14 transcripts were excluded as they were representing non-protein coding sequences or pseudogenes. Additional 26 genes, associated with mitotic cell cycle functions, and further 17 genes related to cell proliferation, adhesion and differentiation,

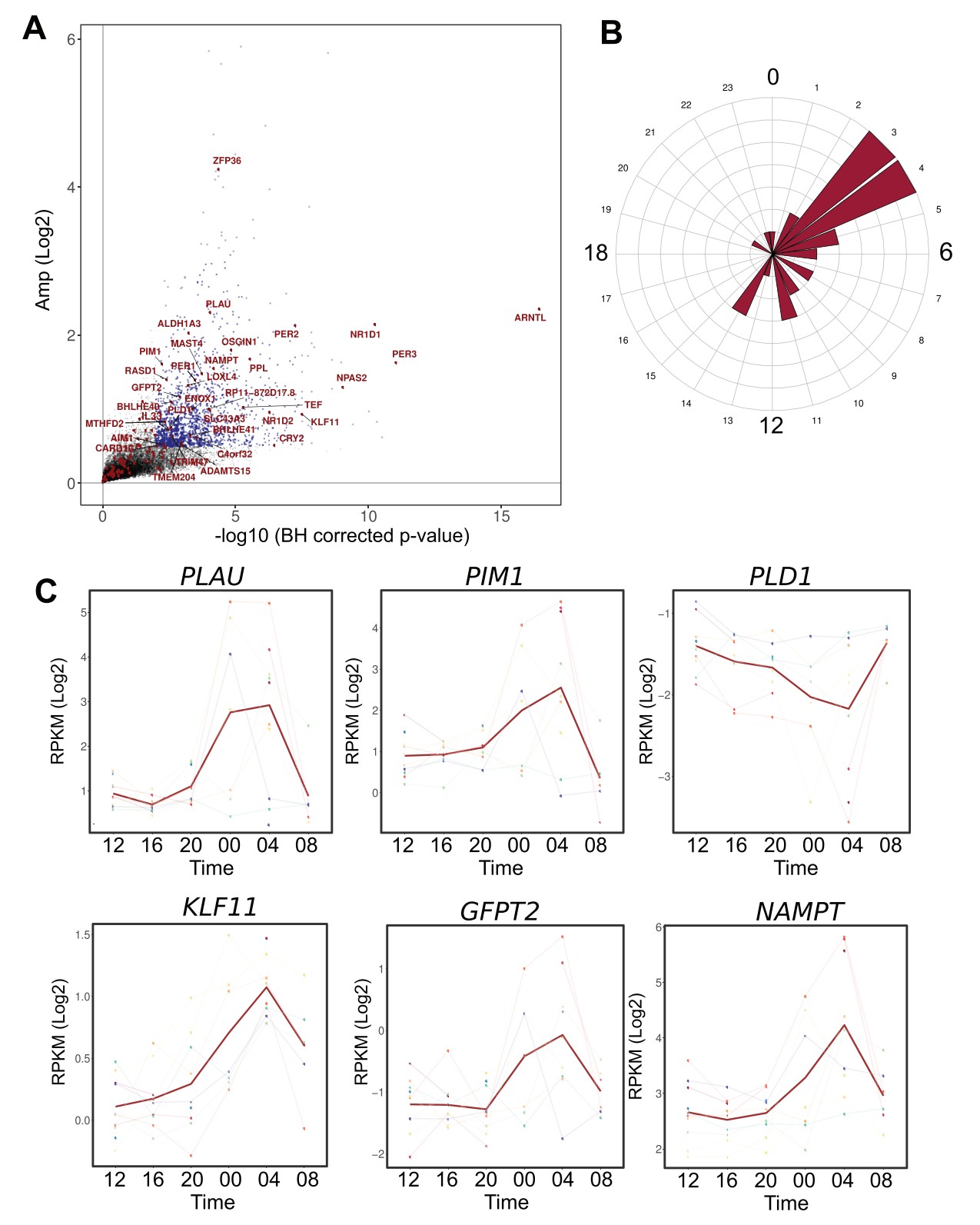

**Figure 5.** Comparison between the circadian transcriptome of human skeletal muscle and human primary myotubes. (**A**) Scatter plot, representing the amplitude of expression in relation to the corrected p-value for genes that were rhythmic in vivo (human muscle biopsies). Genes that were also rhythmic in vitro (hSKM, models 1–4) are colored in red. Blue dots represent genes with a p-value <0.01 and log$_2$ amp >0.5. (**B**) Phase distribution plot

*Figure 5 continued on next page*

*Figure 5 continued*

of genes rhythmic in muscle biopsies and primary myotubes shows enrichment at the 04:00 time point. (**C**) Examples of genes, involved in glucose homeostasis and muscle regeneration, that are rhythmic in vivo and in vitro (RNA-seq data, N = 10).

DOI: https://doi.org/10.7554/eLife.34114.022

The following source data is available for figure 5:

**Source data 1.** List of 190 genes, rhythmic in hSKMs.

DOI: https://doi.org/10.7554/eLife.34114.023

were only found in hSKM and are likely a consequence of incomplete myotube differentiation in cell culture, as opposed to fully differentiated muscle tissue. Cultured muscle cells lack the in vivo environment and the chemical communication that exists within the tissue. Notably, the absence of neuronal connections further limits the final differentiation of cultured myotubes (for review see [*Aas et al., 2013*]).

Among the remaining 133 genes, 99 were expressed in human muscle biopsies. Within this group of overlapping genes, 35 were also qualified as rhythmic at the mRNA level (q-val <0.05) in muscle tissue (*Figure 5—source data 1*). The genes rhythmic in both in vivo and in vitro datasets included the core clock components (*Figure 5A*), as well as functional genes that were enriched at the 04:00 time point (*Figure 5B*). Interestingly, genes implicated in glucose homeostasis (*KLF11*, *GFPT2*, *NAMPT*) and in muscle regeneration (*PLAU*, *PLD1*, *PIM1*), were rhythmic both in vivo and in vitro, suggesting an important role for the circadian clock in regulating muscle physiology (*Figure 5C*).

## Discussion

This study presents the first large-scale circadian transcriptome RNA-seq analysis in muscle biopsies from multiple volunteers and in hSKM cells synchronized in vitro with 2 hr resolution over 48 hr, in the presence of a functional or disrupted cell-autonomous clock, with subsequent analysis of its impact on gene expression. Moreover, we demonstrate that *CLOCK* depletion in cultured primary skeletal myotubes led to significant changes in gene expression (*Figure 2*), and related physiological outputs, comprising the regulation of basal and insulin-stimulated glucose uptake, lipid homeostasis (*Figure 3*), and myokine secretion, as summarized in *Figure 6*. These results provide new insights into the targets of the molecular clock in human skeletal muscle, previously only studied in rodents (*McCarthy et al., 2007*; *Miller et al., 2007*; *Dyar et al., 2014*; *Zhang et al., 2014*; *Dyar et al., 2015*). Finally, to dissect the effects of the cell-autonomous endogenous clock from SCN signals and entrainment cues, this dataset was compared to the diurnal transcriptome of human skeletal muscle collected in form of serial muscle biopsies across 24 hr (*Figures 1* and *5*).

### Comparison between the circadian transcriptome of synchronized myotube cells in vitro, and human muscle tissue collected in vivo

Our in vitro myotube system allows us to explore the transcriptional regulation of muscle target genes without confounding effects of the SCN, rest-activity and feeding-fasting cycles (*Harfmann et al., 2015*). Regarding the rhythmic analysis of in vitro RNA-seq data, larger variations were observed between the two donors than between siControl and siCLOCK conditions (*Figure 4— figure supplement 1*), likely due to the genetic heterogeneity among human individuals. The low number of subjects therefore represents a limitation of our study, despite high time-resolution of 2 hr for sample collection conducted over 48 hr, that resulted in as many as 25 time points per myotube donor. We therefore concentrated on genes, which were rhythmic in both donors in siControl condition, irrespectively of their rhythmicity disruption by siCLOCK treatment. In total, 994 circadian genes (7.65% of the global transcriptome) were rhythmic in at least one of the four models (models 1–4, *Figure 4A–B*), exceeding the value found for U2OS cells, exhibiting 1.5% of oscillating gene transcripts (*Krishnaiah et al., 2017*). When compared with the diurnal transcriptome of human skeletal muscle biopsies, the percentage of rhythmic genes was considerably lower, likely due to the cell culture situation where the circadian amplitude is gradually lost (*Figure 2—figure supplement 2B*) in the absence of entrainment (*Hughes et al., 2009*), or due to effects driven by the SCN or behavioral rhythms rather than by the local peripheral clock. Moreover, we cannot exclude that

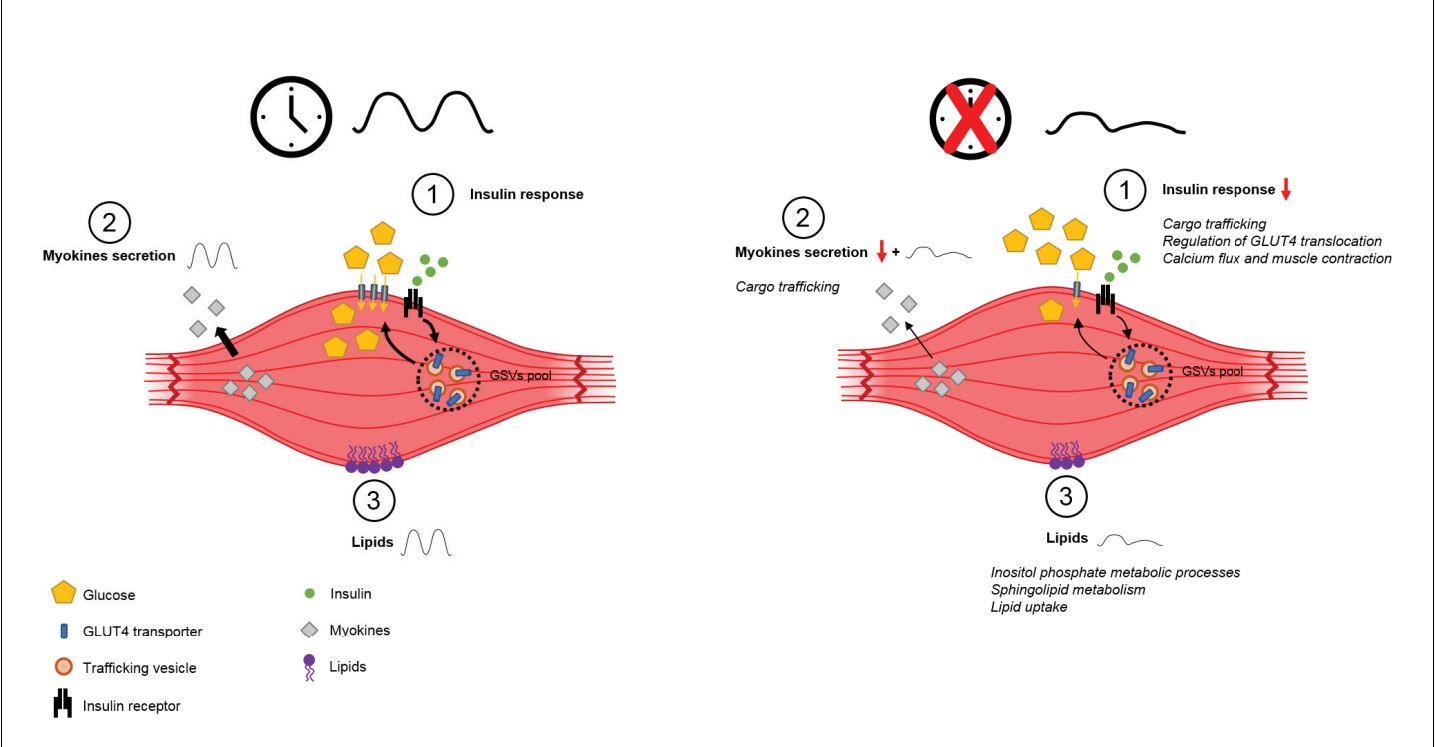

**Figure 6.** Schema summarizing impact of clock disruption on muscle metabolic function. Clock disruption leads to impaired insulin sensitivity and decrease in glucose uptake (1), causes a dysregulation of genes involved in vesicle trafficking (2) and impacts lipid metabolism and lipid metabolite oscillations (3) as reported in *Loizides-Mangold et al. (2017)*.

DOI: https://doi.org/10.7554/eLife.34114.024

discrepancies between the in vivo and in vitro circadian datasets are in part also influenced by the fiber type composition of *vastus lateralis* and *gluteus maximus,* as demonstrated by myosin isoform analysis (*Loizides-Mangold et al., 2017*). Additional differences with respect to gene rhythmicity between the two datasets may stem from potential differences due to the different algorithms employed for the data analyses.

Among the 35 genes classified as rhythmic in cell culture and in skeletal muscle tissue, were genes involved in glucose metabolism and in muscle regeneration, including *PLAU* (*LluisLluís et al., 2001*) and *PIM1* kinase (*Fischer et al., 2009*), along with core clock components such as *NR1D1* and *NR1D2* (*Figure 5A*), previously identified as the most rhythmic transcripts across all human and mouse datasets (*Laing et al., 2015*). Phospholipase *PLD1* (*Teng et al., 2015*), involved in intracellular membrane trafficking and maintenance of glucose homeostasis in human skeletal muscle (*Huang et al., 2005*) was rhythmically expressed in vivo and in vitro. Moreover, oscillatory genes in both datasets included *NAMPT, KLF11,* and *GFPT2*, the latter controlling the flux of glucose into the hexosamine pathway, tightly linked to hyperglycemia and insulin resistance (*Coomer and Essop, 2014*). The expression of *NAMPT*, a key regulator of $NAD^+$ synthesis and muscle maintenance (*Frederick et al., 2016*), was previously shown to be regulated by CLOCK and BMAL1 in complex with SIRT1 (*Ramsey et al., 2009*; *Garten et al., 2015*). Importantly, the diurnal rhythm of secreted NAMPT is disturbed by sleep loss, and positively associates with impairment of postprandial glucose metabolism (*Benedict et al., 2012*). The transcription factor *KLF11*, a glucose-inducible regulator of insulin transcription and secretion, that is a member of the Krüppel-like family of transcription factors proposed as circadian (*Yoshitane et al., 2014*), was found to be regulated by the circadian clock in mouse kidney and epididymal fat tissue (*Guillaumond et al., 2010*), and is possibly involved in postprandial glucose metabolism in skeletal muscle (*Neve et al., 2005*).

Comparison of our in vitro dataset to the results published on U2OS cells (*Hughes et al., 2009*; *Krishnaiah et al., 2017*) revealed that among the 190 genes that were rhythmic in human skeletal

myotubes, 30 were also rhythmic in U2OS. Among those were members of the core-clock machinery, and multiple genes involved in cell cycle progression and mitosis. Moreover, the sulfatase *ARSJ*, involved in glycosphingolipid metabolism, *EXOSC8*, a regulator of mRNA stability, *LRRC16A*, involved in actin filament organization, and *TUBA1C*, encoding for a structural constituent of the cytoskeleton, as well as *E2F1* were rhythmic in skeletal myotubes and in U2OS cells. Interestingly, the RING finger domain protein-encoding gene *TRIM47* exhibited a rhythmic expression in human skeletal myotubes, in human muscle biopsies, and in U2OS cells.

A comparison of our in vivo dataset (exonic signals) to published temporal gene expression databases of mouse skeletal muscle (*Dyar et al., 2014*; *Zhang et al., 2014*) revealed 107 common rhythmic genes between mouse and human skeletal muscle. When comparing the circadian phases of core-clock components in our database to the temporal profiles of the corresponding genes in rodents (*Dyar et al., 2014*; *Zhang et al., 2014*), we observed a phase shift of 8–10 hr. This result is in good agreement with a phase shift observed between peripheral clocks in nocturnal versus diurnal species, which is indeed typically smaller than 12 hr (*Mure et al., 2018*). The question remains at what level such a phase-shift between nocturnal and diurnal species occurs, and why it is not exactly 12 hr in peripheral organs (*Mure et al., 2018*). Further comparative studies, conducted in the same type of tissue and with the same methodology, will be required to explore this fundamental issue.

An important observation was the strong induction of genes associated with inflammation and immune response in human muscle in the early morning hours (04:00) (*Figure 1F*), 16 hr after sampling of the first biopsy. We cannot fully rule out that repeated muscle sampling contributed to this signature, as previously reported for repeated biopsy sampling of a single muscle via the same skin incision site over 25 hr (*Friedmann-Bette et al., 2012*). However, clinical sampling was optimized to minimize this effect, as serial *vastus lateralis* biopsies were sampled across alternating limbs and from separate skin incision sites, each proximal to the previous (*Van Thienen et al., 2014*), not excluding the possibility that circulating molecules may diffuse an inflammatory signal between limbs (*Catoire et al., 2012*). Importantly, this immune signature was restricted to a single time point in the early morning hours, and thus likely does not exclusively result from responses to muscle injury, which would have further increased at the last time point (08:00). Given that inflammatory cytokines have been described as myokines and important regulators of muscle physiology, this could thus represent a true signature with relevant outcomes for muscle physiology.

One limitation of the comparison between RNA-seq datasets obtained for in vivo and in vitro skeletal muscle and hSKM samples in the present study stems from different analyses methods applied for the two datasets. The algorithms applied here for these two datasets were chosen to optimally fit each dataset differing in the number of time points and the time span of samples collection. Indeed, cellular samples were collected every 2 hr over 48 hr resulting in 25 time points, whereas muscle tissue biopsies were collected every 4 hr over 24 hr, resulting in six time points only, due to obvious practical limitations of repetitive muscle tissue biopsy sampling from the same individuals.

## Effect of CLOCK depletion on myotube gene transcription and core clock gene regulation

Efficient clock disruption in adult hSKM cells via siRNA-mediated *CLOCK* knockdown by our previously validated protocol led to significant changes in gene expression (*Figure 2A and B*) (*Perrin et al., 2015*; *Petrenko et al., 2016*; *Loizides-Mangold et al., 2017*). Most core clock genes were downregulated upon siCLOCK transfection, in addition to a flattening of the *Bmal1*-luciferase profile (*Figure 2—figure supplement 2B*), consistent with our previous data for hSKM and human pancreatic islets (*Perrin et al., 2015*; *Saini et al., 2016*). However, despite the observed amplitude blunting, core clock components still presented remnant circadian expression profiles that can likely be attributed to the partial downregulation of *CLOCK,* and to compensatory mechanisms that occur to maintain the circadian machinery (*DeBruyne et al., 2007*) (*Figure 2A*, *Figure 2—figure supplement 2A*), leading to the observation that the effect on absolute gene expression was more pronounced than the effect on mRNA rhythmicity (*Figure 4C and D*). Although our established experimental system for cellular clock disruption mediated by efficient *CLOCK* knockdown proved highly useful to study transcriptional and functional outputs in cultured human primary cells, one should keep in mind that core-clock genes also perform clock unrelated functions. The same holds true for genetic mouse models, where different core-clock gene KO strains exhibit distinct

phenotypes. To discriminate between clock-related and unrelated effects of the *CLOCK* gene knock-down, alternative methods for the circadian clock perturbation will be required.

## Muscle fiber type parameters are affected in response to siCLOCK

*Gluteus maximus* is a slow muscle characterized by high levels of *MYH7* expression, fatigue resistance, and slow speed contraction as well as an oxidative metabolic type (*Talbot and Maves, 2016*). Although the levels of key transcription factors regulating fiber-type-specific genes, including *MYOD1*, *NFATC1*, *SIRT1* and *PPARGC1A* were not significantly altered upon siCLOCK, we identified an upregulation of multiple genes characteristic for type I slow fibers, as well as a downregulation of genes associated with type II fast fibers (*Supplementary file 1*-table S7). Whether these changes affect mitochondrial activity stays to be further explored. Importantly, a recent study has demonstrated that *Bmal1* KO myotubes display reduced mitochondrial respiration and a reduced expression of hipoxia-inducible factor 1 (HIF1) target genes (*Peek et al., 2017*). In agreement, we also observed reduced expression of the HIF target gene *VEGFA* upon clock disruption (*Supplementary file 1*-table S7), supporting the hypothesis that clock - HIF interactions play an important role in the glycolytic capacity of skeletal muscle. In addition, we also observed an upregulation of myosin light chain kinase *MLCK* (*MYLK*) that contributes to force generation by myofilaments. Taken together these observations reinforce the hypothesis that clock disruption induces a global switch in the genetic program towards slow type I muscle fibers, as it was previously suggested in muscle-specific *Bmal1* KO mice (*Hodge et al., 2015*).

## Muscle clock alteration impairs glucose uptake in response to insulin

Skeletal muscle is responsible for 70–80% of insulin-stimulated glucose uptake (*DeFronzo and Tripathy, 2009*). Importantly, we observed a 30% decrease in glucose uptake for both basal and insulin stimulated conditions in siCLOCK-transfected hSKM (*Figure 3C*). Previous studies have reported similar observations in either *Bmal1*-specific muscle KO, or in *ClockΔ19* mutant mice (*Kennaway et al., 2007*; *Dyar et al., 2014*; *Harfmann et al., 2016*). Recently, it was shown that cardiomyocyte-specific *Bmal1* KO and *ClockΔ19* mutant mice exhibit defects in insulin-regulated processes, including over-activation of AKT and mTOR signaling (*McGinnis et al., 2017*). Although we did not see significant changes in *GLUT1* or *GLUT4* gene expression levels, our differential analysis highlighted many genes involved in the regulation of the GLUT4 translocation pathway (*Supplementary file 1*-table S2 and *Figure 2—source data 1*).

Upon closer analysis of the GLUT4 translocation and recycling pathways, we observed changes upon siCLOCK treatment at each step, with several genes being differentially expressed. Specifically, the enzyme PI4K2A, catalyzing the phosphorylation of phosphatidylinositol (PI) to phosphatidylinositol 4-phosphate (PI4P), was downregulated at the mRNA level, which may result in decreased PIP2 and PIP3 levels (*Pullen et al., 1998*; *Sakamoto and Holman, 2008*). Additionally, siCLOCK-depleted cells overexpressed *MAPKAP1* (*mSIN1*), one component of the mTORC2 complex required for AKT phosphorylation (*Frias et al., 2006*), and *CAV-3*, essential for PI3K and AKT activity as well as GLUT4 translocation in muscle (*Fecchi et al., 2006*; *Tan et al., 2012*). Moreover, the observed reduction of *14-3-3θ* (*YWAHQ*) upon siCLOCK at both the mRNA and protein level may lead to an attenuated inhibition of TBC1D1 and TBC1D4 (AS160), and thus block GLUT4 translocation to the plasma membrane (*Ramm et al., 2006*; *Roach et al., 2007*; *An et al., 2010*; *Kleppe et al., 2011*; *Szekeres et al., 2012*). Consistent with this theory, we observed a modest upregulation of the Rab-GTPase-activator *TBC1D1*, in addition to a downregulation of *RGC2* and an upregulation of *TPM3* at the mRNA level. In summary, regulation of GLUT4 translocation and recycling pathways may be affected upon clock disruption with important consequences on glucose uptake and insulin sensitivity as summarized in *Figure 6*.

## Muscle clock disruption influences the expression of genes involved in vesicle trafficking

GLUT4 located at the plasma membrane, is endocytosed in clathrin-coated vesicles and further recycled (*Leto and Saltiel, 2012*; *Jaldin-Fincati et al., 2017*). We observed that several factors of the clathrin-mediated endocytosis machinery were altered upon *CLOCK* depletion (*Supplementary file 1*-table S2), among them *FNBP1* (*FBP17*), *EPN2, HIP1, and SYT1*. Furthermore,

our results suggest that *CLOCK* depletion impacts on calcium levels in the cytoplasm as SYT1 acts as a calcium sensor, and in the presence of elevated $Ca^{2+}$ levels promotes the fusion of close membranes (*Martin, 2015*).

Once GLUT4 is endocytosed, it is transported to early endosomes using RAB5 (*Stenmark et al., 1994*; *Leto and Saltiel, 2012*). As *RAB5B* was upregulated upon siCLOCK, it is suggesting that this recycling step might be increased. Moreover, we found downregulation of *TBC1D16*, which was shown to regulate RAB4 activity, suggesting a possible increase in the fast remobilization of GLUT4 at the plasma membrane (*Goueli et al., 2012*).

We have previously demonstrated that the basal secretion of myokines, such as IL-6, IL-8, and MCP-1, exhibits a circadian pattern, which was strongly disrupted in hSKM after *CLOCK* silencing in vitro (*Perrin et al., 2015*). Here, our transcriptional analysis showed that key regulators of the exocytosis machinery were altered upon clock disruption (*Supplementary file 1*-tables S2 and S3). Both *VAMP3* and *STX6*, which are involved in IL-6 secretion in mouse macrophages (*Manderson et al., 2007*), were downregulated at the mRNA level (*Supplementary file 1*-tables S2 and S6), confirming previous results that clock disruption impacts on vesicle trafficking and secretion (*Marcheva et al., 2010*; *Saini et al., 2016*). Importantly, when compared with results from clock disrupted human islets (*Saini et al., 2016*) we found that numerous genes involved in hormone secretion by pancreatic islets were affected in a similar manner in hSKM (*Supplementary file 1*-table S8).

Further downstream, GLUT4 is sent to the late endosome for degradation by the lysosome or targeted to the endosomal recycling compartment (ERC), through its interaction with VAMP3 (*Dugani et al., 2008*; *Rose et al., 2009*). PI4K2A, which was downregulated upon clock depletion (*Supplementary file 1*-table S2), might be involved here as it regulates VAMP3 trafficking to perinuclear membranes (*Volchuk et al., 1995*; *JovicJović et al., 2014*). In addition, *CAMSAP2*, involved in microtubule stabilization (*Hendershott and Vale, 2014*), and *KIF13A*, associated with recycling endosome tubules (*Delevoye et al., 2014*), were also downregulated upon *CLOCK* disruption (*Supplementary file 1*-tables S2 and S3). Taken together these results, as summarized in *Figure 6*, suggest that the muscle clock may play an important regulatory function in the control of the secretion machinery via transcriptional regulation.

## Cell-autonomous clock disruption in hSKM might impact energy substrate utilization

The circadian clock has been associated with the control of muscle development and regeneration, as clock mutant mice exhibit defects in muscle metabolism, architecture and composition (for review see [*Chatterjee and Ma, 2016*; *Schiaffino et al., 2016*]). Here, we found alterations in the expression of several genes involved in lipid metabolism, calcium handling, electron transport chain, and glucose metabolism (*Figure 2C–E*, *Supplementary file 1*-table S7), suggesting a shift in energy substrate utilization upon clock disruption, as has been proposed previously in rodents upon loss of *Bmal1* (*Hodge et al., 2015*; *Harfmann et al., 2016*). AMP-activated protein kinase, a potent regulator of skeletal muscle fat metabolism (*Thomson and Winder, 2009*) might be dysregulated upon clock disruption as we observed upregulation of its regulatory subunit PRKAG2 and downregulation of subunit PRKAG3. Previous work reported downregulation of both subunits in *Bmal1*-specific muscle KO mice (*Hodge et al., 2015*), suggesting that this gene could be directly controlled by BMAL1.

Clock disruption causes changes in lipid levels as has been described previously for the liver of *Per1/2* KO mice (*Adamovich et al., 2014*). In hSKM, siCLOCK treatment affected several genes involved in lipid metabolic processes, lipid storage and transport (*Figure 2C–E*), which resulted in total phosphatidylcholine and glycosylceramide level alterations (*Figure 2F*). Specifically, we found an increase in the long chain fatty acid transporter *CD36* and in *FABP3*, consistent with previous results obtained in mouse skeletal muscle upon clock disruption (*Hodge et al., 2015*; *Schiaffino et al., 2016*). In addition, we observed an upregulation of *MSTN* upon siCLOCK, which could further promote the increase in *CD36* and *FABP3*, leading to impaired glucose uptake (*Figure 3D*). Interestingly, muscle-specific *myostatin* (*Mstn*) KO mice exhibit a reduction of lipid transporters, including FABP3 and CD36, a diminution of lipid oxidation, higher levels of saturated and unsaturated fatty acids, and a decrease of cardiolipin and triglycerides (*Baati et al., 2017*). Furthermore, downregulation of *Mstn* in skeletal muscle from type one diabetic mice leads to an increase of *Glut1* mRNA and GLUT4 protein levels, promoting insulin-stimulated glucose uptake (*Coleman et al., 2016*). Altogether, these results confirm previous rodents studies and indicate a

shift in substrate utilization in skeletal muscle from carbohydrates to lipids with impact on muscle metabolism and glucose homeostasis (*Dyar et al., 2014*; *Dyar et al., 2015*; *Hodge et al., 2015*; *Harfmann et al., 2016*).

## Conclusions

In summary, our study provides (1) a comparison between rhythmic transcriptome databases obtained from human muscle tissue and cultured primary cells derived from muscle biopsies, and (2) the identification of pathways regulated by *CLOCK* in skeletal muscle, involved in glucose uptake, myokine secretion, and lipid metabolism (*Figure 6*). Human primary cells cultured in vitro have been used as a model to study human disease and metabolism (*Aas et al., 2013*; *Saini et al., 2015*). In combination with tissue analysis as presented here, primary cell culture constitutes a powerful model to study human metabolism, and warrants further analyses in additional metabolically active tissues in physiological conditions, and in the context of metabolic diseases.

# Materials and methods

**Key resources table**

| Reagent type (species) or resource | Designation | Source or reference | Identifiers | Additional information |
|---|---|---|---|---|
| Antibody | anti-AS160 (C69A7) Rabbit mAb | Cell Signaling | Cat. #2670 RRID:AB_2199375 | 1:1000; for western blot; primary Ab |
| Antibody | anti-14-3-3 θ Rabbit polyclonal | Cell Signaling | Cat. #9638 RRID:AB_2218251 | 1:200; for western blot; primary Ab |
| Antibody | anti-CLOCK(H276) Rabit polyclonal | Santa Cruz Biotechnology | Cat. sc-25361 RRID:AB_2260802 | 1:200; for western blot; primary Ab |
| Antibody | anti-actin Rabbit polyclonal | Sigma-Aldrich | Cat. A2066 RRID:AB_476693 | 1:1000; for western blot; primary Ab |
| Antibody | goat anti-rabbit-IgG HRP | Sigma-Aldrich | Cat. A8275 RRID:AB_258382 | 1:3000; for western blot; secondary Ab |
| Recombinant DNA reagent | Bmal1-luciferase (luc) reporter | *Liu et al., 2008*; PMID:18454201 | | |
| Sequence-based reagent | ON-TARGETplus Non-targeting Pool | Dharmacon | D-001810-10-20 | |
| Sequence-based reagent | ON-TARGETplus human CLOCK siRNA SMARTpool | Dharmacon | L-008212-00-0020 | Target Sequences: CAACUUGCACCUAUAAAUA CGACAGGACUGGAAACCUA GAACAACGGACACGCAUGA CUAGAAAGAUGGACAAAUC |
| Peptide, recombinant protein | NA | NA | NA | NA |
| Commercial assay or kit | SuperSignal West Pico Chemiluminescent Substrate | Thermo Fisher Scientific | Prod. #34080 | |
| Commercial assay or kit | Quant-iTª RiboGreenª RNA Assay Kit | Thermo Fisher Scientific | R11491 | |
| Commercial assay or kit | RNeasy Mini kit | Qiagen | Ref # 74104 | |
| Commercial assay or kit | TruSeq Stranded Total RNA Library Prep Kit with Ribo-Zero Gold Set A (48 samples, 12 indexes) | Illumina | RS-122–2301 | |
| Commercial assay or kit | TruSeq Stranded Total RNA Library Prep Kit with Ribo-Zero Gold Set B (48 samples, 12 indexes) Indexes only | Illumina | RS-122–2302 | |
| Commercial assay or kit | TruSeq RNA Library Prep Kit v2 | Illumina | RS-122–2001/RS-122–2002 | |

*Continued on next page*

*Continued*

| Reagent type (species) or resource | Designation | Source or reference | Identifiers | Additional information |
|---|---|---|---|---|
| Commercial assay or kit | HiSeq PE Cluster Kit V4 - cBot | Illumina | PE-401–4001 | |
| Commercial assay or kit | HiSeq SBS Kit V4 250 cycle kit | Illumina | FC-401–4003 | |
| Commercial assay or kit | KAPA HiFi HotStart ReadyMixPCR Kit | Kapa BioSystems (Roche) | KK2602 | |
| Commercial assay or kit | Quant-iT PicoGreen dsDNA Assay Kit | Thermo Fisher Scientific | P7589 | |
| Commercial assay or kit | LabChip DNA High Sensitivity Reagent Kit | Perkin Elmer | CLS760672 | |
| Chemical compound, drug | Forskolin | Sigma-Aldrich | F6886 | |
| Chemical compound, drug | Luciferin | Prolume LTD | #260150 | |
| Chemical compound, drug | 2-deoxy-[$^3$H]-D-glucose | PerkinElmer | NET328A001MC | Specific Activity: 5–10 Ci (185-370GBq)/mmol, 1mCi (37MBq) |
| | Insulin | Sigma-Aldrich | I9278 | |
| Chemical compound, drug | Potassium phosphate monobasic | Sigma-Aldrich | P5655 | |
| Chemical compound, drug | HiPerFect transfection reagent | Qiagen | Cat No./ID: 301705 | |
| Chemical compound, drug | Tert-butyl methyl ether | Sigma-Aldrich | #20256 | |
| Chemical compound, drug | Methylamine solution | Sigma-Aldrich | #534102 | |
| Chemical compound, drug | Methanol LC-MS CHROMASOLV | Fluka (Thermo Fisher Scientific) | #34966 | |
| Chemical compound, drug | Water LC-MS CHROMASOLV | Fluka (Thermo Fisher Scientific) | #39253 | |
| Chemical compound, drug | Chloroform, stabilized with ethanol, for HPLC | ACROS Organics (Thermo Fisher Scientific) | #390760010 | |
| Chemical compound, drug | 12:0 PC (DLPC) | Avanti Polar Lipids | #850335 | |
| Chemical compound, drug | 17:0-14:1 PE | Avanti Polar Lipids | LM1104 | |
| Chemical compound, drug | 17:0-14:1 PI | Avanti Polar Lipids | LM1504 | |
| Chemical compound, drug | 17:0-14:1 PS | Avanti Polar Lipids | LM1304 | |
| Chemical compound, drug | 12:0 SM (d18:1/12:0) | Avanti Polar Lipids | #860583 | |
| Chemical compound, drug | C17 Ceramide (d18:1/17:0) | Avanti Polar Lipids | #860517 | |
| Chemical compound, drug | C8 Glucosyl(ß) Ceramide (d18:1/8:0) | Avanti Polar Lipids | #860540 | |
| Software, algorithm | Rstudio | Rstudio | R version 3.3.1 | |
| Software, algorithm | Prism 5 | GraphPad | NA | |
| Software, algorithm | Excel 2016 | Microsoft | NA | |

*Continued on next page*

*Continued*

| Reagent type (species) or resource | Designation | Source or reference | Identifiers | Additional information |
|---|---|---|---|---|
| Software, algorithm | LumiCycle | Actimetrics | NA | |
| Software, algorithm | STAR: ultrafast universal RNA-seq aligner | *Dobin et al., 2013* PMID: 23104886 | | |
| Software, algorithm | edgeR package | *Robinson et al., 2010* PMID: 19910308 | edgeR version 3.16.5 | |
| Software, algorithm | lme4 R package | *Bates et al., 2015* DOI: 10.18637/jss.v067.i01 | | |
| Software, algorithm | lmtest R package | *Zeileis et al., 2002* DOI: 10.18637/jss.v007.i02 | | |
| Software, algorithm | TopGO R package | https://bioconductor.riken. jp/packages/3.3/bioc/vignettes/ topGO/inst/doc/topGO.pdf | | |
| Software, algorithm | GEMTools | http://gemtools. github.io/ | GEMTools v1.7.1 | |
| | Lipid data analyzer II | IGB-TUG Graz University; PMID:21169379 | LDA v.2.5.1 | |
| Software, algorithm | Image Lab | Bio-Rad | | |
| Other | RIPA buffer | Sigma-Aldrich | Cat# R0278 | |

## Human skeletal muscle biopsies

10 healthy volunteers were recruited for the in vivo study (see *Supplementary file 1*-table S1 for donor characteristics). One week prior to the scheduled laboratory visit, participants had to adhere to a consistent daily feeding and sleeping routine. Participants arrived in the laboratory at 19:00 hr on the day prior to the testing day and ingested one standardized meal that first evening. Participants remained for the duration of their stay in a semi-recumbent position. During the waking hours of the testing day, they were given mixed-macronutrient meal-replacement solutions at hourly intervals (Resource, Nestlé, Switzerland) to ensure energy balance. The laboratory was free from natural light and with artificial lighting standardized to 800 lux in the direction of gaze, ambient temperature maintained between 20 and 25°C and noise levels tightly regulated. Participants were not permitted to sleep during waking hours when lights were on (i.e. 07:00-22:00 hr) and wore eye masks whilst trying to sleep during lights-out (i.e. 22:00-07:00 hr). Anesthetic administration (1% lidocaine w/o epinephrine) and skin/fascia incisions for this procedure (*Bergstrom, 1962*) were completed within the hour prior to sleep such that night-time samples could be acquired with minimal discomfort. Six 100 mg biopsy samples were acquired from the *vastus lateralis* muscle at 4 hr intervals (12:00, 16:00, 20:00, 24:00, 04:00 and 08:00 hr) and immediately snap frozen under liquid nitrogen. Samples were taken from each leg in alternating and ascending order with skin incisions separated by 2–3 cm. The study was conducted in accordance with the Declaration of Helsinki and with the approval of the Health Research Authority (NRES Committee South West; 14/SW/0123) and the Swiss Commission cantonal (Canton Vaud) d'éthique de la recherche (Cer-VD). For further details see *Loizides-Mangold et al. (2017)*.

## Human muscle RNA-sequencing and data analysis

Total Stranded RNA-Seq (in vivo muscle samples): RNA was quantified with Ribogreen (Life Technologies, Carlsbad, CA) and quality was assessed on a Fragment Analyzer (Advances Analytical). Sequencing libraries were prepared from 250 ng total RNA using the TruSeq Stranded Total LT Sample Prep Kit (Illumina, San Diego, CA) with the Ribo-Zero Gold depletion set. The procedure was automated on a Sciclone NGS Workstation (Perkin Elmer, Waltham, MA). The manufacturer's protocol was followed, except for the PCR amplification step. The latter was run for 13 cycles with the KAPA HiFi HotStart ReadyMix (Kapa BioSystems, Roche, Switzerland). This optimal PCR cycle

number was evaluated using the Cycler Correction Factor method as described previously (*Atger et al., 2015*). Purified libraries were quantified with Picogreen (Life Technologies) and the size pattern was controlled with the DNA High Sensitivity Reagent kit on a LabChip GX (Perkin Elmer). Libraries were then pooled by 24, and each pool was clustered at a concentration of 8 pmol on 8 lanes of v4 paired-end sequencing flow cells (Illumina). Sequencing was performed for $2 \times 125$ cycles on a HiSeq 2500 strictly following Illumina's recommendations.

Paired-end reads were mapped to the *Homo sapiens* genome (GRCh38/hg38) using *STAR* (*Dobin et al., 2013*) with parameters "–alignIntronMin 20 –alignIntronMax 1000000 –GTF (option –sjdbGTFfile). Mapped reads were quantified in intronic and exonic regions. For each gene, we defined a gene body by merging all the respective transcripts using *BEDtools* (*Quinlan and Hall, 2010*). A region was defined as exonic if it occurs in a least one of the transcripts while an intronic region had to be shared between all the transcripts. We assigned uniquely mapping paired-reads to exonic regions (exon/exon and complete exon reads) or intronic regions (intron-exon and complete intron reads) considering reads orientation. Genes with less than two intronic reads or 10 exonic reads on average were discarded. Intronic and exonic reads count were normalized using edgeR (*Robinson et al., 2010*) by the library scaling factor from the exonic regions and the respective intronic and exonic length (*rpkm()*). Genes with less than $-2$ RPKM (log2) at the exonic level were discarded. Genes with less than $-3$ RPKM (log2) at the intronic level were reported as NA for the intronic quantification.

Rhythmicity was assessed with a linear mixed-effects model using *lmer()* function from the *lme4* R package applied on the log2 normalized reads count. A standard harmonic regression with a 24 hr period was fitted with a donor-dependent random effect on the baseline:

$$y_{ID, t} \sim \cos\left(\frac{2\pi}{24} t\right) + \sin\left(\frac{2\pi}{24} t\right) + (1|ID)$$

where $y_{ID,t}$ is the log2 normalized reads count for patient *ID* at time *t*. This full model was compared to the null model (without the harmonic terms) using the likelihood ratio test function *ltest()* from the lmtest R package. The p-values were computed from a chi-squared distribution and were adjusted using the Benjamini-Hochberg procedure.

Gene ontology analysis was performed using the TopGO R package . Enrichment analysis for GO terms derived from 'Biological Process' was performed for the genes rhythmic in the three groups (R-I.RE, R-I, and R-E) and in the different phase bins. GO terms with p-value <0.05, a minimum number of 3 genes, and less than 200 annotated genes were considered.

## Transcription factors enrichment analysis

Predictions of transcription factor binding sites (MotEvo) and promoter regions were downloaded from http://swissregulon.unibas.ch/sr/downloads (database *Homo sapiens*, hg19:FANTOM5) (*Pachkov et al., 2013*). Transcription factor binding sites were assigned to their corresponding genes using the promoter regions table. Genes rhythmic at the intronic level, and with amplitude larger than 0.5 (log2), were grouped according to their phase in 4 hr bins. All the genes expressed in the dataset were used as a background and a hypergeometric test was computed for the over-representation of transcription factor binding sites in the different bins. Transcription factor binding sites with -log10(p-value) $>10^4$ and belonging to at least five genes were reported and annotated in *Figure 1I*.

## Study participants, skeletal muscle tissue sampling and primary cell culture

Biopsies from the *Gluteus maximus* muscle were derived from donors with their informed consent (see *Figure 2—figure supplement 1* and *Supplementary file 1*-table S1 for donor characteristics). The experimental protocol ('DIOMEDE') was approved by the Ethical Committee SUD EST IV (Agreement 12/111) and performed according to the French legislation (Huriet's law). All donors had HbA1c levels inferior to 6.0% and fasting glycemia inferior to 7 mmol/L, were not diagnosed for type 2 diabetes (T2D), neoplasia or chronic inflammatory diseases, and not doing shift work. Primary skeletal myoblasts were purified and differentiated into myotubes according to the previously

described procedure (*Agley et al., 2015*; *Perrin et al., 2015*). After reaching confluence, myoblasts were differentiated into myotubes during 7–10 days in DMEM supplemented with 2% FBS.

## siRNA transfection and lentiviral transduction

Human primary myoblasts were differentiated into myotubes as described above. Cells were transfected 24 hr (RNA-seq) or 72 hr (western blot, glucose uptake) prior to experiment with 20 nM siRNA targeting *CLOCK* (siCLOCK), or with non-targeting siControl (Dharmacon, Lafayette, CO), using HiPerFect transfection reagent (Qiagen, Hilden, Germany) following the manufacturer's protocol. To produce lentiviral particles, *Bmal1*-luciferase (*Liu et al., 2008*) lentivectors were transfected into 293 T cells using the polyethylenimine method (for detailed procedure see [*Pulimeno et al., 2013*]). Myoblasts were transduced with the indicated lentiviral particles with a multiplicity of infection (MOI) = 3 for each, grown to confluence, and subsequently differentiated into myotubes.

## In vitro skeletal myotube synchronization and real-time bioluminescence recording

Primary myotubes were synchronized with 10 µM forskolin (Sigma-Aldrich, St. Louis, MO) for 60 min at 37°C in a cell culture incubator, then the medium was changed to a phenol red-free recording medium containing 100 µM luciferin (Prolume LTD, Pinetop, AZ), and cells were transferred to a 37°C light-tight incubator as previously described by us (*Pulimeno et al., 2013*). Bioluminescence from each dish was continuously monitored using a Hamamatsu photomultiplier tube (PMT) detector assembly. Photon counts were integrated over 1 min intervals. Luminescence traces are shown as detrended data.

## Glucose uptake measurement

Human myotubes treated with siControl or siCLOCK as described before were serum-starved for 3 hr then incubated with 2-deoxy-[$^3$H]-D-glucose for 15 min. Incubations were performed with or without insulin stimulation (1 hr, 100 nM). After incubation, the medium was removed prior to cell lysis in 0.05 M NaOH. Cell content radioactivity was determined by liquid scintillation counting (Perkin Elmer, 2900TR) and protein content was quantified by using the Bradford protein assay. Glucose transport is expressed in pmol/mg.min (*Chanon et al., 2017*).

## Protein analysis

Human myotubes transfected with siControl or siCLOCK for 24 to 72 hr, were lysed in RIPA buffer. Protein extracts (8 µg) were analyzed by SDS-PAGE and immunoblotted to 0.45 µm nitrocellulose membrane or 0.2 µm PVDF membrane using a wet system (Bio-Rad, Hercules, CA) according to the manufacturer's instructions. Membranes were blocked and incubated with primary and secondary antibodies in 5% BSA/TBS-T 0.5% or 5% BSA/TBS-T 0.1%. Primary and secondary antibodies were used at the following dilutions: anti-TBC1D4/AS160 (1/1000, Cell Signaling, Danvers, MA, #2670S), anti-14-3-3θ (1/200, Cell Signaling, #9638S), anti-CLOCK (1/200, Santa Cruz Biotechnology, Santa Cruz, CA, H-276) and anti-ACTIN (1/1000, Sigma-Aldrich, A2066), anti-rabbit-HRP (1:3000, Sigma-Aldrich A8275). Signals were visualized using SuperSignal West Pico Chemiluminescent Substrate (Thermo Fisher Scientific, Waltham, MA). For protein quantification, five donors were analyzed but only the representative western blot result of one donor is shown.

## Lipidomics

The lipidomics analysis was performed as described in *Loizides-Mangold et al. (2017)*. Briefly, human primary myotubes were harvested from one confluent 10 cm dish (~$1.5 \times 10^6$ cells) and lipid extracts were prepared using the MTBE protocol (*Matyash et al., 2008*). Total phosphorus was determined as described in (*Loizides-Mangold et al., 2017*). Phospho- and sphingolipid were analyzed by mass spectrometry using on a TSQ Vantage Triple Stage Quadrupole Mass Spectrometer (Thermo Fisher Scientific) equipped with a robotic nanoflow ion source (Nanomate HD, Advion Biosciences, Ithaca, NY), using multiple reaction monitoring (MRM). Lipid concentrations were calculated relative to the relevant internal standards and then normalized to the total phosphate content of each total lipid extract. Triacylglyceride analysis was performed by mass spectrometry analysis on a hybrid ion trap LTQ-Orbitrap XL mass spectrometer (Thermo Fisher Scientific) equipped with a

micro LC binary pump UFLC-XR (Shimadzu, Kyoto, Japan). Lipid identification was carried out with the Lipid Data Analyzer II (LDA v. 2.5.1, IGB-TUG Graz University) (*Hartler et al., 2011*).

## hSKM mRNA extraction and quantitative PCR analysis

Differentiated myotubes were synchronized by 10 μM forskolin, collected every 2 hr during 48 hr (0–48 hr), deep-frozen in liquid nitrogen and kept at −80°C. Total RNA was prepared using RNeasy Mini kit (Qiagen). 0.5 μg of total RNA was reverse-transcribed using Superscript III reverse transcriptase (Invitrogen) and random hexamers, and PCR-amplified on a LightCycler 480 (Roche). Mean values for each sample were calculated from the technical duplicates of each RT-qPCR analysis, and normalized to the mean of two housekeeping genes (*HPRT* and *9S*), which served as internal controls. Primers used for this study are listed in *Supplementary file 1*-table S9.

## hSKM RNA-sequencing and data analysis

Total RNA was prepared from primary human skeletal myotubes from two donors, transfected either with siControl or siCLOCK, synchronized with a forskolin pulse and collected every 2 hr during 48 hr in duplicates, using RNeasy Mini Kit (Qiagen). Total RNA libraries were prepared from 300 ng of RNA following customary Illumina TruSeq v2 protocols for next generation sequencing. PolyA-selected mRNAs were purified, size-fractioned, and subsequently converted to single-stranded cDNA by random hexamer priming. Following second-strand synthesis, double-stranded cDNAs were blunt-end fragmented and indexed using adapter ligation, after which they were amplified and sequenced according to protocol. RNA libraries were sequenced with a HiSeq2000 sequencer producing 49pb paired-end reads. Standard quality checks for material degradation (Bioanalyzer and TapeStation, Agilent Technologies, Santa Clara, CA) and concentration (Qubit, Life Technologies) were done before and after library construction, ensuring that samples are suitable for sequencing.

Paired-end reads were mapped to the human genome (version GRCh37/hg19) with GEMTools (v1.7.1) using GENCODE v19 as gene annotation. RNA-seq reads were subsequently filtered for correct orientation of the two ends, a minimum mapping quality score of 150 and allowing a maximum of 5 mismatches in both ends. GENCODE annotation v19 was used to assign filtered reads to their corresponding exons and genes. For each gene, we processed the exons from all transcripts, which were quantified by considering only filtered reads as above, in which both ends map to exons of the same gene. The gene counts were incremented non-redundantly, meaning reads overlapping two exons were counted once to the total count sum per gene.

The differential gene expression analysis was performed with the R package edgeR (*Robinson et al., 2010*). First, transcripts expressed lower than three counts per million (CPM) and noninformative (e.g. non-aligned) were filtered. To minimize the log-fold changes between the samples for most genes, a set of scaling factors for the library sizes were estimated with the trimmed mean of M-values (TMM) method (*Robinson and Oshlack, 2010*). The dispersion was estimated with the quantile-adjusted conditional maximum likelihood (qCML) method. Once the dispersion estimates are obtained, we performed the testing procedures for determining differential expression using the exact test (*Robinson and Smyth, 2008*).

Regarding the rhythmic analysis, homemade algorithm was developed to analyze these RNA-seq data. In short, raw data were transformed to $\log_2$ reads per kilobase per million mapped reads (RPKM) as described previously (*Atger et al., 2015*), then only transcripts with $\log_2$ RPKM >0 for each of the fourth conditions (two subjects, siControl or siCLOCK) were kept avoiding big variability for weakly expressed transcripts. The 48 time points of each condition were used to define a local regression function (LOESS). This step allows smoothing the curve and reducing local variability. The function was then used to calculate 10 different measures (maximum and minimum slopes, first and second extremum, minimum-maximum ratio, autocorrelation, measure of scattering, residues on the loess function, residues on a linear function and period). These features were used to classify gene expression patterns in four different groups: rhythmic genes (category 'circadian'), genes that show only one peak at the beginning of the time course (category 'one peak'), linearly (category 'linear') and scatteredly expressed genes (category 'cloud'). The algorithm attributes a probability to each transcript per condition. To be classified in one category, this probability must be the highest value and superior to 0.5 in at least one category. If no probabilities are superior to 0.5 for the four categories, transcripts are grouped into model 16 (non-rhythmic). The 11 major circadian genes,

including *ARNTL* (*BMAL1*), *NR1D1* (*REVERBα*), *NR1D2* (*REVERBβ*), *PER1, PER2, PER3, CRY1, CRY2, NPAS2, TEF* and *BHLHE41*, were selected to train a random forest model. The same number of genes for the three other groups were also integrated in the training dataset. This dataset was then passed to the training algorithm for random forests and gene conditions that were assigned to one of these categories with a high score (0.9) were integrated in the training dataset. This procedure was repeated until 500 curves per group were identified. The last model was kept to classify the whole dataset.1485 curves from 994 transcripts were identified as rhythmic among 12,985 transcripts. Altogether, transcripts were grouped into 16 models.

## Bioluminescence and statistical data analysis

Bioluminescence data were analyzed with the Actimetrics LumiCycle analysis software (Actimetrics LTD, Wilmette, IL). RNA-seq data and qPCR data analysis were performed using RStudio, GraphPad Prism five and Excel 2016. Panther analyses were performed using the PANTHER version 12.0 released on 10.07.2017. Statistical analyses were performed using Student's *t*-test. Differences were considered significant for (*) p-value <0.05, (**) p-value <0.01, and (***) p-value <0.001. Exact p-values and raw data for *Figures 2* and *3* are listed in *Supplementary file 2*.

## Mycoplasma test for primary cultures

Since primary cultures, established from human skeletal muscle tissue biopsies were used in this study, mycoplasma contamination tests were conducted only once for each primary myotube culture. To do so, 100 µl of culture medium were taken 48 hr following the last trypsinization, boiled at 95°C for 5 min, and centrifuged for 10 s at 14,000 rpm. PCR was performed on 5 µl of thus processed samples, using a mix of primers listed in *Supplementary file 1*-table S9. The PCR program was 5 min at 95°C, followed by 30 cycles with 95°C 30 s, 60°C 30 s, 72°C 30 s, and a final elongation at 72°C for 10 min. PCR products were separated on a 1.5% agarose gel.

## Acknowledgements

The authors thank Jacques Philippe and Sylvain Sardy for constructive discussions; Pamela Pulimeno for help with conducting the in vitro around-the-clock sample collection, RNA extractions and RT-qPCR experiments; Svetlana Skarupelova and Camille Saini for help with RNA extractions and Anne-Marie Makhlouf for lentivirus preparation; Luciana Romano and Deborah Beilser for their help with in vitro RNA sequencing (University of Geneva); Marc Moniatte and Jonathan Paz Montoya (PCF, EPFL) for help with the lipid analyses; and Ondine Walter (NIHS) for the ethical and logistical management of human samples. This work was funded by the Sinergia Swiss National Science Foundation (Grant No. CRSII3-154405 to HR, CD, EL), the Swiss National Science Foundation (Grant No. 31003A-166700 (CD), the Fondation Privée de HUG, Fondation Ernst et Lucie Schmidheiny, the Société Académique de Genève (CD) and by the United Kingdom Biotechnology and Biological Sciences Research Council Grant BB/I008470/1 (JDJ).

## Additional information

### Competing interests

Cédric Gobet, Benjamin D Weger, Eugenia Migliavacca, Aline Charpagne, Jerome N Feige: Is a full-time employee of the Nestlé Institute of Health Sciences SA. James A Betts: Has been a consultant for PepsiCo (Quaker) and Kellogg's. Leonidas G Karagounis: Is an employee of Nestec Ltd. The other authors declare that no competing interests exist.

### Funding

| Funder | Grant reference number | Author |
|---|---|---|
| Schweizerischer Nationalfonds zur Förderung der Wissenschaftlichen Forschung | Sinergia Grant No. CRSII3-154405 | Howard Riezman Etienne Lefai Charna Dibner |

| Schweizerischer Nationalfonds zur Förderung der Wissenschaftlichen Forschung | 31003A-166700 | Charna Dibner |
| Fondation Privée HUG | | Emmanouil T Dermitzakis Charna Dibner |
| Fondation Ernst et Lucie Schmidheiny | | Charna Dibner |
| Société Académique de Genève | | Charna Dibner |
| Biotechnology and Biological Sciences Research Council | | Jonathan D Johnston |

The funders had no role in study design, data collection and interpretation, or the decision to submit the work for publication.

## Author contributions
Laurent Perrin, Conceptualization, Data curation, Formal analysis, Validation, Investigation, Visualization, Methodology, Writing—original draft, Writing—review and editing; Ursula Loizides-Mangold, Conceptualization, Data curation, Formal analysis, Supervision, Validation, Investigation, Methodology, Writing—original draft, Project administration, Writing—review and editing; Stéphanie Chanon, Data curation, Investigation, Visualization, Methodology; Cédric Gobet, Resources, Data curation, Software, Formal analysis, Validation, Investigation, Methodology; Nicolas Hulo, Resources, Data curation, Software, Formal analysis, Supervision, Methodology; Laura Isenegger, Data curation, Software, Formal analysis, Validation; Benjamin D Weger, Investigation, Methodology; Eugenia Migliavacca, Data curation, Investigation, Methodology; Aline Charpagne, Data curation, Software, Formal analysis, Methodology; James A Betts, Jean-Philippe Walhin, Iain Templeman, Keith Stokes, Dylan Thompson, Kostas Tsintzas, Data curation, Formal analysis, Investigation, Methodology, Writing—review and editing; Maud Robert, Validation, Investigation, Methodology, Writing—review and editing; Cedric Howald, Resources, Data curation, Software, Formal analysis, Validation; Howard Riezman, Conceptualization, Funding acquisition, Investigation, Writing—review and editing; Jerome N Feige, Jonathan D Johnston, Data curation, Formal analysis, Supervision, Validation, Investigation, Methodology, Writing—review and editing; Leonidas G Karagounis, Conceptualization, Data curation, Formal analysis, Supervision, Validation, Methodology, Writing—review and editing; Emmanouil T Dermitzakis, Conceptualization, Resources, Data curation, Software, Supervision, Validation; Frédéric Gachon, Conceptualization, Data curation, Software, Formal analysis, Supervision, Validation, Investigation, Methodology, Writing—review and editing; Etienne Lefai, Conceptualization, Data curation, Formal analysis, Supervision, Funding acquisition, Validation, Investigation, Methodology, Writing—original draft, Project administration, Writing—review and editing; Charna Dibner, Conceptualization, Supervision, Funding acquisition, Validation, Investigation, Methodology, Writing—original draft, Project administration, Writing—review and editing

## Author ORCIDs
Ursula Loizides-Mangold http://orcid.org/0000-0001-9233-2974
Nicolas Hulo http://orcid.org/0000-0003-2640-636X
Howard Riezman http://orcid.org/0000-0003-4680-9422
Jonathan D Johnston http://orcid.org/0000-0001-8083-9794
Frédéric Gachon https://orcid.org/0000-0002-9279-9707
Charna Dibner http://orcid.org/0000-0002-4188-803X

## Ethics
Human subjects: The in vivo study was conducted in accordance with the Declaration of Helsinki and with approval of the Health Research Authority (NRES Committee South West; 14/SW/0123) and the Swiss Commission Cantonal (Canton Vaud) d'éthique de la recherche (Cer-VD). Muscle biopsies were obtained from donors with their informed consent. The experimental protocol ('DIOMEDE') for the in vitro study was approved by the Ethical Committee SUD EST IV (Agreement 12/111) and performed according to the French legislation (Huriet's law).

Decision letter and Author response
Decision letter https://doi.org/10.7554/eLife.34114.036
Author response https://doi.org/10.7554/eLife.34114.037

## Additional files

### Supplementary files

• Supplementary file 1. Supplementary tables S1-S9.
DOI: https://doi.org/10.7554/eLife.34114.025

• Supplementary file 2. Detailed list of exact p-values and raw data related to *Figures 2* and *3*.
DOI: https://doi.org/10.7554/eLife.34114.026

• Transparent reporting form
DOI: https://doi.org/10.7554/eLife.34114.027

### Major datasets

The following datasets were generated:

| Author(s) | Year | Dataset title | Dataset URL | Database, license, and accessibility information |
|---|---|---|---|---|
| Gobet C | 2017 | Temporal RNA-seq analysis of human skeletal muscle tissue biopsies | https://www.ncbi.nlm.nih.gov/geo/query/acc.cgi?acc=GSE108539 | Publicly available at the NCBI Gene Expression Omnibus (accession no: GSE108539) |
| Howald C | 2018 | Temporal RNA-seq analysis of human skeletal myotubes synchronized in vitro | https://www.ncbi.nlm.nih.gov/geo/query/acc.cgi?acc=GSE109825 | Publicly available at the NCBI Gene Expression Omnibus (accession no: GSE109825) |

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
