## [Decision Letter]

Thank you for submitting your article "Transcriptomic analyses reveal rhythmic and CLOCK–driven pathways in human skeletal muscle" for consideration by *eLife*. Your article has been reviewed by two peer reviewers, and the evaluation has been overseen by a Reviewing Editor and Fiona Watt as the Senior Editor. The following individuals involved in review of your submission has agreed to reveal his identity: Joseph Bass (Reviewer #2).

The reviewers have discussed the reviews with one another and the Reviewing Editor has drafted this decision to help you prepare a revised submission

Summary:

In this manuscript titled 'Transcriptomic analyses reveal rhythmic and CLOCK–driven pathways in human skeletal muscle', the authors performed RNA–seq analysis in muscle tissue biopsies (in vivo) and cultures of synchronized primary differentiated myofibers (in vitro) from human subjects, allowing them to compare cell autonomous and non–cell autonomous rhythms of transcription. Of note, the authors found that rhythmic transcription of genes involved in glucose transport (i.e. GLUT4 regulation), lipid metabolism and immune response were present in both the in vivo and in vitro samples. However, many other genes which were rhythmic in vivo did not appear rhythmic in vitro, suggesting that there are non–cell autonomous factors controlling rhythmic transcription in human muscle. In addition, since total RNA was used, both pre–mRNA and mature mRNAs were quantified. Indeed, the authors identified genes which oscillate at both the transcriptional and post–transcriptional levels. Finally, the dependence of rhythmic transcripts on the core molecular clock was determined using siRNA–mediated knock down of CLOCK in the primary myotube cultures. siCLOCK myotubes displayed reduced expression of many genes which were found to be rhythmic in vitro. The authors also determined that siCLOCK myotubes display reduced insulin–dependent and –independent glucose uptake, suggesting reduced activity of the GLUT4–transport mechanism. Overall, the finding that the circadian clock controls glucose and lipid metabolic gene expression in primary human myofibers is important and provides further evidence of the role of the molecular clock in the regulation of muscle function and glucose homeostasis.

The daily transcriptome analysis of the human muscle biopsies taken at 4 h intervals across 24 h from 10 healthy individuals serve as a valuable and important resource, especially in view of the current attempts to map circadian rhythmicity in humans. In this regard, a more rigorous comparison between available detests of mouse muscle transcriptome (e.g. McCarthy et al., 2007; Miller et al., 2007; Dyar et al., 2014; Zhang et al., 2014; Hodge et al., 2015) with this new data would provide interesting insight regarding the difference in muscle rhythmic gene expression between mice and humans, and likely between nocturnal and diurnal species, respectively.

The major strength of this study lies on the expression data obtained from human healthy individuals in a well–controlled manner, this section by itself is an important resource.

The in vitro experiments primarily corroborate previous studies done with cultured cells (e.g. Krishnaiah et al., 2017) that show dramatic reduction in rhythmic gene expression in culture compared to organs in vivo.

Essential revisions:

1) The in vivo biopsy samples were taken from the vastus lateralis muscle whereas the in vitro samples were from gluteus maximus. Therefore, is it possible that the observed gene expression differences observed between the in vitro and in vivo sample sets are explained by the fact that they are derived from different muscles? One option to address this would be to compare gene expression profiles from biopsies of VL and GM muscles – do they differ in a similar way to the in vitro versus in vivo gene sets? This may beyond the scope here.

2) The authors observed gene expression changes with siCLOCK that support increased mitochondrial oxidative/type I muscle remodeling. This result is consistent with what was observed in RNA–sequencing studies in Bmal1KO mouse muscle (Hodge et al., 2015). However, others have provided evidence for reduced mitochondrial oxidative metabolism in Bmal1KO muscle cells. Therefore, it is possible that the gene signature does not equate with the metabolic phenotype. For instance, gene expression changes may be compensatory rather than causal? In the absence of such analysis, discussion should acknowledge the interest in future functional bioenergetic profiling (e.g., respirometry with mitochondrial fuel substrates).

3) The mathematical model used to detect rhythmicity should be validated on an existing dataset or a demo dataset in order to compare their results with established and widely used algorithms that detect rhythmic profiles, such as JTK_CYCLE (Hughes et al., 2010) and Meta Cycle (Wu et al., 2016) that incorporates ARSER, JTK_CYCLE and Lomb–Scargle.

4) It would be interesting to compare the genes identified as rhythmic in cultured U2OS cells (Krishnaiah et al., 2017) with the ones identified in the current study, even though these are completely different cells. It might shed light on genes, aside from the core clock, that globally maintain rhythmicity in culture.

5) The authors conclusion that the cell–autonomous circadian clock has an essential role in coordinating muscle glucose homeostasis and lipid metabolism in humans should be revised as it is not supported by their results. Their finding that knockdown of clock in vitro affected the overall expression of ~8% of all genes with genes related to glucose and lipid metabolism suggests that the CLOCK protein itself modulate their expression and not necessarily the oscillator. Furthermore, the result that only few genes maintain rhythmicity both in vivo and in vitro, most of them are core clock components, does not support it either.

---

## [Author Response]

Summary:In this manuscript titled 'Transcriptomic analyses reveal rhythmic and CLOCK–driven pathways in human skeletal muscle', the authors performed RNA–seq analysis in muscle tissue biopsies (in vivo) and cultures of synchronized primary differentiated myofibers (in vitro) from human subjects, allowing them to compare cell autonomous and non–cell autonomous rhythms of transcription. Of note, the authors found that rhythmic transcription of genes involved in glucose transport (i.e. GLUT4 regulation), lipid metabolism and immune response were present in both the in vivo and in vitro samples. However, many other genes which were rhythmic in vivo did not appear rhythmic in vitro, suggesting that there are non–cell autonomous factors controlling rhythmic transcription in human muscle. In addition, since total RNA was used, both pre–mRNA and mature mRNAs were quantified. Indeed, the authors identified genes which oscillate at both the transcriptional and post–transcriptional levels. Finally, the dependence of rhythmic transcripts on the core molecular clock was determined using siRNA–mediated knock down of CLOCK in the primary myotube cultures. siCLOCK myotubes displayed reduced expression of many genes which were found to be rhythmicin vitro. The authors also determined that siCLOCK myotubes display reduced insulin–dependent and –independent glucose uptake, suggesting reduced activity of the GLUT4–transport mechanism. Overall, the finding that the circadian clock controls glucose and lipid metabolic gene expression in primary human myofibers is important and provides further evidence of the role of the molecular clock in the regulation of muscle function and glucose homeostasis.The daily transcriptome analysis of the human muscle biopsies taken at 4 h intervals across 24 h from 10 healthy individuals serve as a valuable and important resource, especially in view of the current attempts to map circadian rhythmicity in humans. In this regard, a more rigorous comparison between available detests of mouse muscle transcriptome (e.g. McCarthy et al., 2007; Miller et al., 2007; Dyar et al., 2014; Zhang et al., 2014; Hodge et al., 2015) with this new data would provide interesting insight regarding the difference in muscle rhythmic gene expression between mice and humans, and likely between nocturnal and diurnal species, respectively.

We thank the reviewers for this insightful comment as indeed a comparison to the published mouse databases was lacking. Following the reviewers’ suggestion, we first compared genes rhythmic at the exonic level in human skeletal muscle based on our in vivo RNAseq data analysis to previously published transcriptional data performed on mouse skeletal muscle biopsies. We chose for comparison the datasets published by Zhang and colleagues (Zhang et al., 2014) representing temporal RNAseq and microarray analysis conducted in mouse skeletal muscle under a constant darkness regimen, and the work by Dyar and colleagues (Dyar et al., 2014) where the analysis was conducted by microarray under a 12h light–dark regimen. It should be noted that our study was done in human skeletal muscle biopsies obtained from a different muscle type as compared to the published rodent studies (vastus lateralis in our study, versus gastrocnemius, soleus and tibia anterior in the above–mentioned mouse studies). As shown by Venn diagram in Author response image 1, 107 genes were qualified as rhythmic in all three databases.

Zooming in on the specific differences between human and mouse data in skeletal muscle, we found that in human skeletal muscle rhythmic transcription was distributed into two sharp phases of transcript accumulation with the majority of genes peaking at 04:00, and a second smaller peak appearing at 16:00. This bimodal pattern of gene expression has been observed for rodents as well, where the majority of genes peak at CT23 (Zhang et al., 2014) or CT18 (McCarthy et al., 2007; Miller et al., 2007), and minor peaks of enrichment were observed at CT8 or CT8–10, respectively. These data suggest that an overall bimodal expression profile might be kept between rodents and humans, with the major peak time corresponding to late night – early morning hours, prior to the light onset. Keeping in mind that mice are nocturnal animals and humans are diurnal, this major transcriptional peak seems to correspond to the end of the active phase in mice and to the end of the rest phase in humans. The meaning of this difference stays to be explored but we cannot exclude that our experimental procedures also contributed to the observed difference.

With regard to specific genes, we observed in humans a peak of expression for genes related to muscle contraction and mitochondrial activity during the active phase (Supplemental dataset 2). For nocturnal animals, such as rodents, rhythmic genes related to muscle contraction however, do not exclusively peak during the active phase at night, but cluster into two peaks that correspond to the active and the resting phases (McCarthy et al., 2007; Zhang et al., 2014). McCarthy and colleagues report that two genes (*Myh3, Actc1*) related to muscle contraction peak at CT2 and one gene (*Myh10*) is enriched at CT14. According to CIRCA (Zhang et al., 2014), *Myh2, Myh6*, and *Myh7* peak during the resting phase in mice, and *Myh9* peaks towards the end of the active phase. Further differences were observed for genes involved in lipid metabolism. *Elov5*, which was enriched at the end of the resting phase in humans (04:00), as shown in our recent manuscript (Loizides–Mangold et al., 2017), reached its peak of expression in rodents at the end of the active period (CT22–24, (Zhang et al., 2014; Hodge et al., 2015)). Taken together these results suggest that in rodent skeletal muscle, rhythmic transcriptional activity might be differently regulated compared to human skeletal muscle with regard gene expression at the active/absorptive phase or resting phase.

Whereas the circadian phase for core–clock genes is kept between diurnal and nocturnal species at the SCN level, a phase–shift is observed in peripheral organs. Curiously, the observed shift in peripheral clocks among diurnal and nocturnal species is typically smaller than 12 hours and corresponds to rather 8–10 hours (Mure et al., 2018). In good agreement with previous studies, when comparing the circadian phases of core–clock genes in our database to the temporal profiles of the corresponding rodent data (Dyar et al., 2014; Zhang et al., 2014), we observed a similar phase shift of 8–10 hours. The question remains at what level such a phaseshift between nocturnal and diurnal species occurs, and why it is not exactly 12 h in peripheral organs (Mure et al., 2018). Further comparative studies, conducted in the same type of tissue and the same methodology, will be required to explore this fundamental issue.

**Author response image 1. respfig1:** Comparison of our in vivo vastus lateralis rhythmic dataset with mouse skeletal muscle rhythmic data (Dyar et al., 2014; Zhang et al., 2014). Blue = our in vivo RNAseq data; red = (Dyar et al., 2014) soleus; green = (Zhang et al., 2014). The numbers presented in the Venn diagram correspond to the total number of rhythmically expressed genes for each segment (exons only). For the human in vivo dataset (our work), this corresponds to R–E + R–I.R–E, with the cutoff described in the manuscript (Figure 1). For (Dyar et al., 2014) (microarray) and (Zhang et al., 2014) (microarray) rhythmic genes were identified using a Benjamini–Hochberg q–value <0.2 in the JTK_Cycle algorithm.

**Author response image 2. respfig2:** 

We therefore revised the corresponding sentence in the Results and Discussion sections as follows:

Results section: "As previously reported in mice, rhythmic gene transcription was distributed into two phases of transcript accumulation (04:00 and 16:00, Figure 1E). The afternoon peak (16:00) was enriched in genes related to muscle contraction and mitochondrial activity (Figure 1–source data 2) whereas homologous genes in rodents are shared between the active but also the resting phase (McCarthy et al., 2007; Miller et al., 2007; Hodge et al., 2015).”

Discussion section: “On the other hand, a comparison of our in vivo dataset (exonic signals) to published temporal gene expression databases of mouse skeletal muscle (Dyar et al., 2014; Zhang et al., 2014) revealed 107 common rhythmic genes between mouse and human skeletal muscle. When comparing the circadian phases of core–clock components in our database to the temporal profiles of the corresponding genes in rodents (Dyar et al., 2014; Zhang et al., 2014), we observed a phase shift of 8 – 10 hr. This result is in good agreement with a phase shift observed between peripheral clocks in nocturnal versus diurnal species, which is indeed typically smaller than 12 h (Mure et al., 2018). The question remains at what level such a phase–shift between nocturnal and diurnal species occurs, and why it is not exactly 12 h in peripheral organs (Mure et al., 2018). Further comparative studies, conducted in the same type of tissue and with the same methodology, will be required to explore this fundamental issue.”

The major strength of this study lies on the expression data obtained from human healthy individuals in a well–controlled manner, this section by itself is an important resource.The in vitro experiments primarily corroborate previous studies done with cultured cells (e.g. Krishnaiah et al., 2017) that show dramatic reduction in rhythmic gene expression in culture compared to organs in vivo.Essential revisions:1) The in vivo biopsy samples were taken from the vastus lateralis muscle whereas the in vitro samples were from gluteus maximus. Therefore, is it possible that the observed gene expression differences observed between the in vitro and in vivo sample sets are explained by the fact that they are derived from different muscles? One option to address this would be to compare gene expression profiles from biopsies of VL and GM muscles – do they differ in a similar way to the in vitro versus in vivo gene sets? This may beyond the scope here.

We entirely agree with the reviewers that the muscle origin might have contributed to the observed gene expression differences. To address this important point, we have previously quantified the expression of myosin isoforms *MYH1, MYH2, MYH4* and *MYH7* in cDNA samples obtained from human gluteus maximus and from vastus lateralis biopsies (Loizides–Mangold et al., 2017). As expected, significant differences in myosin isoform composition between vastus lateralis and gluteus maximus were observed. The dominant myosin isoform in vastus lateralis was *MYH2* followed by low levels of *MYH1* and *MYH7*, indicative of mixed fast 2A/2X fibers. The dominant isoforms for gluteus maximus were *MYH2* and *MYH7*, followed by *MYH1* indicative of mixed slow and fast 2A fibers. The expression of *MYH4*, specific for the fastest 2B fiber type, was very low (Harrison et al., 2011).

To further emphasize this valuable point made by the reviewers, we have added the following sentences to the Discussion section:

“Moreover, we cannot exclude that discrepancies between the in vivoand in vitro circadian datasets are in part also influenced by the fiber type composition of *vastus lateralis* and *gluteus maximus,* as demonstrated by myosin isoform analysis (Loizides–Mangold et al., 2017).”

To further dissect the impact of the skeletal muscle source on skeletal muscle gene expression, we compared the number of genes commonly expressed in our in vitro and in vivo RNAseq datasets, with published genomic data on human biopsies from *quadriceps* and *gluteus maximus* muscle (Eisenberg et al., 2008)., We detected that the number of genes differentially expressed between our in vivo and in vitro datasets is substantially higher than the number of genes differentially expressed between the two biopsies originating from gluteus maximus and quadriceps. Indeed, in the published dataset (Eisenberg et al., 2008) a total of 954 genes (875+79) were differentially expressed. This number was significantly lower compared to the number of genes differentially expressed between skeletal myotubes derived from gluteus maximus tissue biopsies and vastus lateralis tissue. However, 875 of these genes were shared between the published data, and the genes that were differentially expressed in vitro and in vivo in our datasets comparison (gluteus in vitro vs vastus lateralis in vivo). This overlap is significant and cannot be explained by a random overlap between two datasets–. That means that there is an effect of the muscle type origin on the transcriptomic profile of gluteus maximus in vitro vs vastus lateralis in vivo. Given the overwhelming number of genes differentially expressed between gluteus maximus (in vivo) and vastus lateralis (in vitro), it is tempting to speculate that the in vitro transition has an even stronger impact on gene expression than the origin of the muscle tissue biopsy. However, the effect of the muscle type on gene expression is significant, and should be taken into account when interpreting the data.

2) The authors observed gene expression changes with siCLOCK that support increased mitochondrial oxidative/type I muscle remodeling. This result is consistent with what was observed in RNA–sequencing studies in Bmal1KO mouse muscle (Hodge et al., 2015). However, others have provided evidence for reduced mitochondrial oxidative metabolism in Bmal1KO muscle cells. Therefore, it is possible that the gene signature does not equate with the metabolic phenotype. For instance, gene expression changes may be compensatory rather than causal? In the absence of such analysis, discussion should acknowledge the interest in future functional bioenergetic profiling (e.g., respirometry with mitochondrial fuel substrates).

Indeed, a recent publication by the Bass group (Peek et al., 2017) has shown that disruption of *Bmal1* in skeletal myotubes and in fibroblasts reduces mitochondrial respiration and anaerobic glycolysis through interaction of the circadian clock with the Hypoxia Inducible Factor (HIF) pathway. This finding indicates that rhythmic oxygen levels can reset circadian clocks, something that has been also suggested by the Asher lab (Adamovich et al., 2017). We agree with the reviewers that it cannot be excluded that the observed increase in slow twitch muscle gene expression (Supplementary file 1–table S7), does not necessarily equate to increased mitochondrial activity. Moreover, in agreement with the data published by Peek et al., we also observe a reduction in the expression of the HIF target gene *VEGFA* upon clock disruption (Supplementary file 1), supporting the hypothesis that clock – HIF interactions play an important role in the glycolytic capacity of skeletal muscle.

Following this constructive remark, we have now modified the corresponding paragraph in the Discussion section, emphasizing the impact of oxygen sensing on skeletal muscle function in addition to muscle type I remodeling:

"[…]we identified an upregulation of multiple genes characteristic for type I slow fibers, as well as a downregulation of genes associated with type II fast fibers (Supplementary file 1). Whether these changes affect mitochondrial activity stays to be further explored. Importantly, a recent study has demonstrated that *Bmal1* KO myotubes display reduced mitochondrial respiration and a reduced expression of Hipoxia Inducible Factor (HIF) target genes (Peek et al., 2017). In agreement, we also observed reduced expression of the HIF target gene *VEGFA* upon clock disruption (Supplementary file 1), supporting the hypothesis that clock – HIF interactions play an important role in the glycolytic capacity of skeletal muscle."

3) The mathematical model used to detect rhythmicity should be validated on an existing dataset or a demo dataset in order to compare their results with established and widely used algorithms that detect rhythmic profiles, such as JTK_CYCLE (Hughes et al., 2010) and Meta Cycle (Wu et al., 2016) that incorporates ARSER, JTK_CYCLE and Lomb–Scargle.

The well–known harmonic regression (COSINOR, (Nelson et al., 1979) was used to assess rhythmicity of the in vivo RNA–seq data. In order to deal with the fact that time points are repeated measures from the same patient, we used a mixed linear model. This includes the usual fixed effects from the harmonic regression model (or COSINOR) and a random effect (patient–specific) on the intercept that deals with the subject–to–subject variation and dependency of the repeated measures. This allows to, not violating the assumption of independent error terms behind the linear regression theory. A similar method has been used to characterize rhythmic transcript in human blood samples (Arnardottir et al., 2014).

Following the pertinent suggestion by the reviewers regarding the rhythmic analysis of the in vitro samples, we evaluated the temporal expression profiles of key core clock genes, such as *ARNTL, NR1D1, NR1D2, PER1, PER2, PER3, CRY1, CRY2, NPAS2, TEF* and *BHLHE41*, extracted from two published large–scale time series (Atger et al., 2015; Petrenko et al., 2017). The expression profiles of all genes listed above, extracted from these two databases, were identified by the algorithm developed and employed by us for the in vitroanalysis as circadian.

We further evaluated our in vitroalgorithm on the model dataset cycMouseLiverRNA from the MetaCycle R package. 20 core–clock and functional genes, listed in Revision table 1, were classified as circadian by our algorithm. The algorithm maps time series gene expression to one of the 4 categories: CIRCA: circadian; LIN: linear; ONE: one peak followed by a flat expression profile; NUA: scattered (from nuage, French word for cloud). The algorithm provides a probability for each category, with the gene expression profile being assigned to the category with the highest probability, presented in the right–most column.

Revision Table 1. Evaluation of core–clock and functional genes from the model dataset cycMouseLiverRNA by our in vitro algorithm. Note that all the genes qualified as circadian by the MetaCycle R package were also confirmed by our analysis method.

NameCIRCALINNUAONEPredictionAvpr1a_1418603_at0.86200.0860.052CIRCACirbp_1416332_at0.8400.0840.076CIRCAClock_1418659_at0.8600.0860.054CIRCAElovl5_1437211_x_at0.86200.0860.052CIRCAFkbp5_1448231_at0.86600.0860.048CIRCAHist1h1c_1416101_a_at0.86800.0860.046CIRCAHsp90aa1_1437497_a_at0.7640.0060.1920.038CIRCALipg_1421262_at0.79200.0760.132CIRCANr1d1_1426464_at0.8600.0860.054CIRCANr1d2_1416958_at0.8600.0860.054CIRCANr1h3_1450444_a_at0.86400.10.036CIRCAPer1_1449851_at0.87600.080.044CIRCAPer2_1417602_at0.82400.0760.1CIRCARgs16_1426037_a_at0.86600.0860.048CIRCARorc_1425792_a_at0.8600.0860.054CIRCAScap_1433520_at0.86200.0860.052CIRCATef_1424175_at0.8600.0860.054CIRCATsc22d3_1420772_a_at0.86200.0860.052CIRCATspan4_1448276_at0.8600.0860.054CIRCATubb5_1416256_a_at0.86200.0860.052CIRCA

Of note, the commonly used JTK_CYCLE algorithm (Hughes et al., 2010) did not prove reliable for our data set, most probably due to the large amplitude differences between the two circadian cycles, which were observed in our experimental settings. A high number of genes qualified as “circadian” by JTK_CYCLE algorithm, were clearly false–positives. The example profiles of the gene expression profiles qualified as circadian by JTK_CYCLE are presented in Author response image 3.

**Author response image 3. respfig3:** Examples of the gene expression profiles from our database qualified as circadian by JTK_CYCLE.

4) It would be interesting to compare the genes identified as rhythmic in cultured U2OS cells (Krishnaiah et al., 2017) with the ones identified in the current study, even though these are completely different cells. It might shed light on genes, aside from the core clock, that globally maintain rhythmicity in culture.

We thank the reviewers for this constructive comment and have subsequently compared our dataset to the results published for U2OS cells by (Krishnaiah et al., 2017). Among the 190 genes that were rhythmic in human skeletal myotubes, 30 were also rhythmic in U2OS cells. Among those were members of the core–clock machinery such as *ARNTL, CRY1–2, NR2D1–2, TEF, PER1–3*, and multiple genes involved in cell cycle progression and mitosis. Beyond those also the sulfatase *ARSJ*, involved in glycosphingolipid metabolism, *EXOSC8*, a regulator of mRNA stability, *LRRC16A*, involved in actin filament organization, and *TUBA1C*, a structural constituent of the cytoskeleton, as well as *E2F1* were rhythmic in both skeletal myotubes and U2OS cells. Interestingly, the RING finger domain protein–encoding gene *TRIM47* exhibited a circadian expression in human skeletal myotubes, in human muscle biopsies, and in U2OS cells. The candidate genes that we highlighted in our study for being highly relevant to muscle physiology, and which exhibited circadian profiles in both synchronized myotubes and in skeletal muscle biopsies (*KLF11, GFPT2, NAMPT, PLAU, PLD1, PIM1*), were all significantly expressed in U2OS cells. However, those genes did not exhibit rhythmic profiles in U2OS cells. *KLF11* had a rhythmic tendency, although it did not reach statistical significance to qualify as circadian according to JTK–cycle (adjusted p–value of 0.266437).

To incorporate this information the following part was added to the Discussion section:

“Comparison of our in vitro dataset to the results published on U2OS cells (Hughes et al., 2009; Krishnaiah et al., 2017) revealed that among the 190 genes that were rhythmic in human skeletal myotubes, 30 were also rhythmic in U2OS. Among those were members of the core–clock machinery, and multiple genes involved in cell cycle progression and mitosis. Moreover, the sulfatase *ARSJ*, involved in glycosphingolipid metabolism, *EXOSC8*, a regulator of mRNA stability, *LRRC16A*, involved in actin filament organization, and *TUBA1C*, encoding for a structural constituent of the cytoskeleton, as well as *E2F1* were rhythmic in skeletal myotubes and in U2OS cells. Interestingly, the RING finger domain protein–encoding gene *TRIM47* exhibited a rhythmic expression in human skeletal myotubes, in human muscle biopsies, and in U2OS cells.

5) The authors conclusion that the cell–autonomous circadian clock has an essential role in coordinating muscle glucose homeostasis and lipid metabolism in humans should be revised as it is not supported by their results. Their finding that knockdown of clock in vitro affected the overall expression of ~8% of all genes with genes related to glucose and lipid metabolism suggests that the CLOCK protein itself modulate their expression and not necessarily the oscillator. Furthermore, the result that only few genes maintain rhythmicity both in vivo and in vitro, most of them are core clock components, does not support it either.

The authors thank the reviewers for this insightful commentary. Indeed, the current interpretation of the in vitro results might be misleading. We subsequently changed the respective parts in the Discussion section and toned down the message regarding the importance of the cell–autonomous regulation of human skeletal muscle and highlighting the distinction between circadian clock–related and unrelated effects of *CLOCK* gene knockdown. The revised paragraphs read as following:

Abstract: “Our findings suggest an essential role for the circadian coordination of skeletal muscle glucose homeostasis and lipid metabolism in humans.”

Discussion section: “Moreover, we demonstrate that *CLOCK* depletion in cultured primary skeletal myotubes led to significant changes in gene expression (Figure 2), and related physiological outputs, comprising the regulation of basal and insulin stimulated glucose uptake, lipid homeostasis (Figure 3), and myokine secretion, as summarized in Figure 6.”

Discussion section: “Though our established experimental system for cellular clock disruption mediated by efficient *CLOCK* knockdown proved highly useful to study transcriptional and functional outputs in cultured human primary cells, one should keep in mind that core–clock genes also perform clock unrelated functions. The same holds true for genetic mouse models, where different core–clock gene KO strains exhibit distinct phenotypes. To discriminate between clock–related and unrelated effects of the *CLOCK* gene knockdown, alternative methods for the circadian clock perturbation will be required.”